



# The role of upper ocean heat content in the regional variability of Arctic sea ice at sub-seasonal time scales

Elena Bianco[1,2], Doroteaciro Iovino[1], Simona Masina[1], Stefano Materia[3,4], and Paolo Ruggieri[5]

[1]Ocean modeling and Data Assimilation Division, Centro Euro-Mediterraneo Sui Cambiamenti Climatici (CMCC), Bologna, Italy
[2]Department of Environmental Sciences, Informatics and Statistics, Ca' Foscari University, Venice, Italy
[3]Climate Simulations and Predictions Division, Centro Euro-Mediterraneo Sui Cambiamenti Climatici (CMCC), Bologna, Italy
[4]Barcelona Supercomputing Center, Barcelona, Spain
[5]Department of Physics and Astronomy, University of Bologna, Bologna, Italy

**Correspondence:** Elena Bianco (elena.bianco@cmcc.it)

**Abstract.** In recent decades, the Arctic Ocean has undergone changes associated with enhanced poleward inflow of Atlantic and Pacific waters and increased heat flux exchange with the atmosphere in seasonally ice-free regions. The associated changes in upper ocean heat content can alter the exchange of energy at the ocean-ice interface. Yet, the role of ocean heat content in modulating Arctic sea ice variability is poorly documented, particularly at regional scale. We analyze ocean heat transports

and surface heat fluxes between 1980-2021 using two eddy-permitting global ocean reanalyses, C-GLORSv5 and ORAS5, to assess the surface energy budget of the Arctic Ocean and its regional seas. We then assess the role of upper ocean heat content, computed in the surface mixed layer ($Q_{ml}$) and in the 0-300 m layer ($Q_{300}$), as a sub-seasonal precursor of sea ice variability by means of lag correlations. Our results reveal that in the Pacific Arctic regions, sea ice variability in autumn is linked with $Q_{ml}$ anomalies leading by 1 to 3 months, and this relationship has strengthened in the Laptev and East Siberian seas during 2001-

2021 relative to 1980-2000, primarily due to reduced surface heat loss since the mid-2000s. $Q_{300}$ anomalies act as a precursor for wintertime sea ice variability in the Barents and Kara seas, with considerable strengthening and expansion of this link from 1980-2000 and 2001-2021 in both reanalyses. Our results highlight the role played by upper ocean heat content in modulating the interannual variability of Arctic sea ice at sub-seasonal timescales. Heat stored in the ocean has important implications for the predictability of sea ice, calling for improvements in forecast initialization and focus upon regional predictions in the Arctic

region.

## 1 Introduction

Satellite observations have documented a rapid decline of Arctic sea ice in all seasons (Simmonds, 2015; Stroeve and Notz, 2018; Onarheim et al., 2018). The mean sea ice state is transitioning to a new regime of thinner, more fractured and mobile ice (e.g., Graham et al., 2019; Sumata et al., 2023). This results in increased vulnerability to dynamic and thermodynamic

forcing mechanisms, including pulses of ocean heat from the lower latitudes (Holland et al., 2006), temperature fluctuations (Olonscheck et al., 2019) and enhanced ocean mixing and ocean-ice heat fluxes (e.g., Duarte et al., 2020; Ricker et al., 2021).



The interplay between the atmosphere, ocean and sea ice is central to explaining the pronounced temporal and spatial variability of the Arctic ice cover observed at a range of time scales (e.g., England et al., 2019; Ding et al., 2019). These complex interactions pose a challenge for accurate sea ice predictions, particularly at the regional level.

Sea ice variability in the Arctic is strongly region-dependent (e.g., Onarheim et al., 2018). The Atlantic sector and in particular the Barents Sea have experienced larger sea ice loss than other areas of the Arctic Ocean (Onarheim et al., 2014; Lind et al., 2018), due to enhanced heat advection by the warm and saline Atlantic Water (AW) entering the basin at intermediate depths (200-800m, Aagaard 1989). Heat from the Atlantic Ocean reaches the Arctic through the Barents Sea Opening (Smedsrud et al., 2010) and Fram Strait (Schauer and Beszczynska-Möller, 2009). The combined effects of increased volume inflow (Smedsrud

et al., 2022) and warmer temperatures (Wang et al., 2019) of the AW current have been linked to sea ice decline in the Barents Sea (e.g., Årthun et al., 2012; Smedsrud et al., 2013) and more recently in the Eastern Eurasian Basin (Polyakov et al., 2020b). In these regions, the interaction between AW and sea ice has also been examined in relation to the weakening of the stratified, cold halocline layer and consequent increase in vertical heat fluxes towards the surface (Polyakov et al., 2010, 2017). These changes are collectively referred to as the 'Atlantification' of the Arctic Ocean (Årthun et al., 2012; Asbjørnsen et al., 2020)

and are thought to be the primary cause of thermodynamic ice melt during winter, when incoming solar radiation is absent (Onarheim et al., 2014; Tsubouchi et al., 2021; Ivanov et al., 2012).

The Pacific sector of the Arctic has also been the epicenter of some of the most remarkable episodes of sea ice loss in recent years (Comiso et al., 2017). While small in comparison to the inflow from Fram Strait (about 10 times smaller in volume and with a heat flux that is 1/3 of the Fram Strait heat flux, Woodgate et al. 2012), heat transport through Bering Strait has been

linked to sea ice variability in the Pacific Arctic (Serreze and Meier, 2019; Wang and Danilov, 2022). The poleward inflow of Pacific Water (PW) has shown an increase since the early 2000s, due to changes in volume transport and a modest increase in observed temperature (Woodgate, 2018). The associated subsurface warming has been shown to accelerate sea ice melt (MacKinnon et al., 2021) and act as a trigger for early melt onset (Woodgate et al., 2010).

Enhanced poleward heat transport from the lower latitudes contributes to altering the Arctic's energy budget, causing the

amplification of upper ocean warming (Shu et al., 2022; Asbjørnsen et al., 2020). Mayer et al. (2019) estimated an energy imbalance of the Arctic Ocean of the order of 1 $\mathrm{W}\,m^{-2}$ between 2001–17, with 1/3 of the accumulated heat going to the sea ice. This finding is corroborated by remarkable upward trends in upper ocean heat content over the past four decades, particularly during summer and autumn (Li et al., 2022). The temperature increase of Arctic Ocean is at the core of the ice-albedo feedback (Perovich et al., 2007; Perovich and Polashenski, 2012). As more areas of open ocean become exposed to

solar radiation due to sea ice break-up and early retreat, the seasonal upper ocean heat uptake increases and a surplus of energy becomes stored in the surface layers (Stammerjohn et al., 2012; Serreze and Meier, 2019), where it is immediately available to the ice through lateral and bottom melting (e.g., Carmack et al., 2015).

To date, studies addressing oceanic drivers of sea ice variability have mainly focused on quantifying the effect of northward heat transport from the Pacific and Atlantic Oceans (e.g., Onarheim et al., 2015; Nummelin et al., 2017; Dörr et al., 2021;

Aylmer et al., 2022). This generally entails a focus on interannual timescales of variability, owing to the delayed response of sea ice to heat anomalies that originate from lower latitudes (e.g., Woodgate et al., 2010). In comparison, the role of





ocean heat content as a precursor of sea ice anomalies on sub-seasonal timescales has been far less explored. An improved characterization of the co-variability of sea ice and upper ocean heat content is of practical value to stakeholder groups, who are primarily interested in short-term predictability. Moreover, the accelerating trends towards a thinner sea ice cover implicate

a higher vulnerability to regional scale forcing (Perovich and Polashenski, 2012).

Additionally, the role of the Arctic surface mixed layer, which represents the link between the ocean, sea ice and atmosphere (Toole et al., 2010), has received little attention in the literature, despite recent assessments of changes in the thermal state of the upper ocean have documented warming trends, especially in regions of maximum sea ice retreat (e.g., Polyakov et al., 2020a). While the relative amounts of upper ocean heat that are lost to the atmosphere and to the overlying sea ice remain

hard to quantify, there is evidence that a surplus of energy absorbed by the mixed layer in summer can act to delay autumn ice growth (Perovich et al., 2007; Steele et al., 2008; Ivanov et al., 2016). However, our understanding of the effects of mixed layer heat on sea ice variability is still limited, partly due to the scarcity of sustained subsurface observations (e.g., Yang, 2006).

In this work, we use two eddy-permitting global ocean reanalyses, the CMCC C-GLORSv5 and ECMWF ORAS5, to quantify changes in the Arctic Ocean heat budget and assess the influence of upper ocean heat content on regional sea ice variability

at sub-seasonal time scales. Ocean reanalyses provide uniformly gridded reconstructions of the state of the ocean climate by combining ocean models driven by atmospheric forcing and available observational data via advanced data assimilation methods (Balmaseda et al., 2015). While they have become an established tool for a variety of climate services and science-driven studies (Storto et al., 2019), ocean reanalyses are especially valuable for the assessment and monitoring of the state of the Arctic Ocean, where hydrographic observations are severely limited by perennial ice cover (e.g., Uotila et al., 2019). Furthermore,

their performance in polar regions has been widely evaluated with encouraging results (e.g., Iovino et al., 2022; Lien et al., 2016; Ilıcak et al., 2016). With respect to fully modeled data, errors and uncertainties in the representation of ocean variables in reanalyses are effectively reduced through the assimilation of observational fields. The products adopted in this study have also been employed in several applications (e.g. Takahashi et al., 2021; Carton et al., 2019) including in polar regions (e.g., Mayer et al., 2019; Shu et al., 2021). However, very few studies using ocean reanalyses have investigated changes in pan-Arctic

heat budget and ocean heat content and, to our knowledge, no previous study has explored these aspects in relation to sea ice variability with a regional focus.

The remainder of this article is structured as follows: Section 2 introduces the data and methodological approach of the study. Section 3 presents the results, which are further divided into three parts: Section 3.1, which provides an overview of the dominant changes in the Arctic Ocean between 1980-2021; Section 3.2, where we perform a heat budget analysis to ensure the

accuracy of our results; and Section 3.3, where we examine the coupling between ocean heat content and sea ice variability in the past 40 years. In Section 4, we discuss our findings and their implications for the future climate and provide a summary in Section 5.



## 2 Data and Methods

### 2.1 Sea ice and ocean reanalyses

Monthly means of sea ice concentrations between 1980 and 2021 are derived from the fifth generation atmospheric global reanalysis of the European Center for Medium-Range Weather Forecasts (ECMWF ERA5, Hersbach et al. 2020). ERA5 data are available from the Copernicus Climate Change Service (C3S) Data Store (Thépaut et al., 2018) on a regular latitude-longitude grid with 0.25° x 0.25° horizontal resolution, from 1950 to present day. Sea ice concentration in the ERA5 data set after 1979 relies on the EUMETSAT Ocean and Sea Ice Application Facilities (OSI-SAF) Climate data Record version

1.2 (OSI-409a) and operational OSI-SAF (OSI-430) products (1979-2007 and 2007-present, respectively), which are based on combined passive microwave observations from the Defense Meteorological Satellite Program (DMSP) Special Sensor Microwave/Imager (SSM/I) and the Special Sensor Microwave Imager/Sounder (SSMIS).

**Table 1.** Overview of Ocean Reanalyses used in this study

| Product Name | Model / DA Scheme | Resolution (horizontal, vertical) | Atmospheric forcing | Observations (SST and SIC) | Reference |
|---|---|---|---|---|---|
| C-GLORSv5 | NEMO 3.2-LIM2 / OceanVAR | 0.25° x 0.25°, 75L | ERA-Interim | NOAA and NSIDC DMSP[2] | Storto and Masina (2016) |
| ORAS5 | NEMO 3.4.1-LIM2 / 3D-Var FGAT | 0.25° x 0.25°, 75L | ERA-Interim/ ECMWF OPS[1] | HadISSTv2 and OSTIA[3] | Zuo et al. (2019) |

[1] up to 2015/ After 2015

[2] SST is assimilated from the NOAA SST 1/4° analyses (Reynolds et al., 2007). With respect to sea ice data assimilation, C-GLORSv5 introduces a nudging scheme to weakly constrain sea ice thickness in the Arctic from PIOMAS.

[3] SST is assimilated from HadISSTv2 pentad before 2008 and from OSTIA analysis from 2008. SIC data come from the OSTIA reprocessed analysis before 2008 and OSTIA analysis after.

We derive temperature, mixed layer depth (MLD), surface heat fluxes and velocity fields at monthly resolution between 1980-2021 from the Global Ocean Reanalysis System version 5 from the Euro-Mediterranean Center on Climate Change (CMCC

C-GLORSv5; Storto and Masina 2016) and the Ocean ReAnalysis System 5 from the European Center for Medium-Range Weather Forecasts (ECMWF ORAS5; Zuo et al. 2019), which is available for download on the C3S Data Store. An overview of the key product characteristics is provided in Table 1. Both C-GLORSv5 and ORAS5 are produced with the NEMO ocean model (Madec et al., 2017) coupled to the Louvain-la-Neuve Sea ice Model (LIM2, Fichefet and Maqueda 1997). Both models



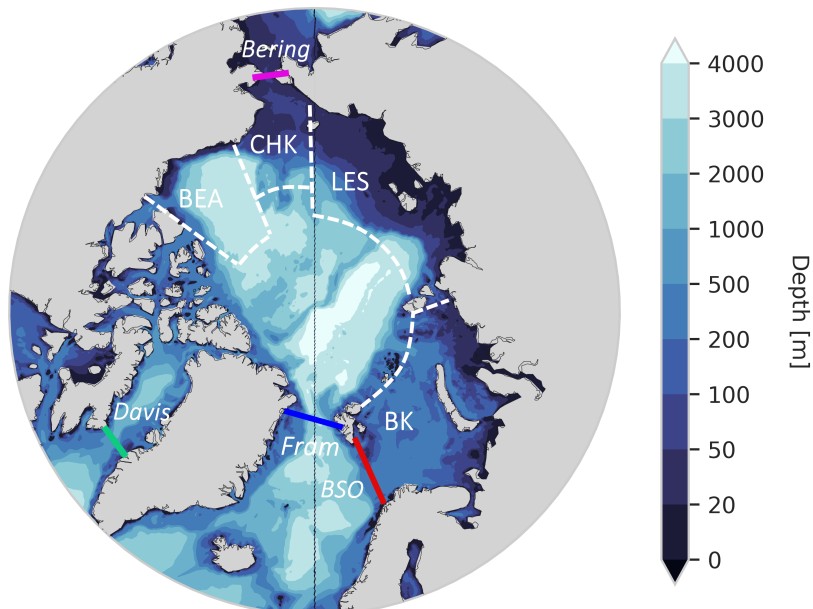

**Figure 1.** Arctic Ocean bathymetry in C-GLORSv5. Dashed lines indicate the boundaries of the regional domains considered in this study (Barents-Kara, BK; Laptev-East Siberian, LES; Chukchi Sea, CHU and Beaufort Sea, BEA). Solid lines mark the location of the main Arctic gateways: Fram Strait, Barents Sea Opening (BSO), Bering Strait and Davis Strait. Note the irregular depth intervals in the colorbar.

adopt a tripolar grid at eddy-permitting configuration with 0.25° x 0.25° horizontal resolution. The original vertical resolution
is 50 and 75 vertical levels for C-GLORSv5 and ORAS5, respectively, however we adopt a version of C-GLORSv5 with 75 vertical levels (C-GLORSv5e025L75, herein referred to as simply C-GLORSv5). In both reanalyses, the ocean is forced by the ECMWF ERA-Interim atmospheric reanalysis (Dee et al., 2011); in ORAS5, this is replaced by ECMWF operational NWP after 2015. The two products differ in their assimilation schemes in terms of input observational datasets, frequency of analysis, assimilation time-windows and bias correction schemes. A full description of the assimilated satellite and in-situ observations
as well as the specifics of data assimilation schemes can be found in the reference papers listed in Table 1.

### 2.2 Regional and pan-Arctic surface heat budget

We first estimate the surface heat budget in the C-GLORSv5 and ORAS5 reanalyses for the whole Arctic and for four sub-domains: the Barents-Kara region (BK); the Laptev-East Siberian region (LES); the Chukchi Sea (CHU) and the Beaufort Sea (BEA, Fig. 1). These regional domains are largely consistent with those of previous studies (e.g., Stroeve et al., 2016; Bliss
et al., 2019; Lenetsky et al., 2021).

Under the assumption of mass and salinity conservation, the ocean's surface heat budget is given by the balance between advective, vertical and diffusive heat flux terms (1):



$$\underbrace{\frac{\partial Q}{\partial t}}_{Qt} = \rho_0 C_p \underbrace{\int_S VT \, dS}_{OHT} + \underbrace{\int_A Qs \, dA}_{SSHF} + Q_{diff} \qquad (1)$$

where $Qt$ is the ocean heat content tendency; $OHT$ represents the advective ocean heat transport through a vertical section $S$, the reference density $\rho_0$ (1026 $\mathrm{kg}\,m^{-3}$) and specific heat capacity $C_p$ (3996 $\mathrm{J}\,kg^{-1}\,K^{-1}$) of sea water are both assumed constant, $V$ and $T$ are the cross-sectional velocity and potential temperature, respectively; SSHF (positive downward) indicates the net sea surface heat flux, $Qs$, over an area $A$; $Qdiff$ represents the diffusive heat transport.

We define the pan-Arctic domain as the area enclosed by the four ocean gateways: the Fram, Bering, and Davis straits and the Barents Sea Opening (BSO) (Fig. 1), as in previous observational (e.g., Tsubouchi et al., 2012) and modeling studies (e.g., Lique and Steele, 2013). The net ocean heat transport into the Arctic is computed along each section (on the native grid) at monthly frequency for C-GLORSv5 and ORAS5 (2)

$$OHT = \rho_0 C_p \int_{-Z(\lambda)}^{\eta} \int_{\lambda 1}^{\lambda 2} VT \, d\lambda dz \qquad (2)$$

Where $\lambda 1$ and $\lambda 2$ are the coordinates of the section line, $Z(\lambda)$ is the depth at each section, and T is taken relative to a reference temperature of 0°C (e.g., Årthun et al., 2012; Lique and Steele, 2013).

We consider the net sea surface heat flux term (SSHF, $\mathrm{W}\,m^{-2}$) over the box area enclosed by the four gates. SSHF is defined positive downward in both reanalyses and represents the exchange of solar and non-solar fluxes between the ocean and atmosphere. Note that in considering the heat budget components in (1) we omit the diffusive heat transport term as this is negligible compared to the advective and heat fluxes terms (see Lique and Steele, 2013). For a detailed analysis of the coupled ocean-ice-atmosphere surface energy budget of the Arctic Ocean, which is beyond the scope of this study, the reader can refer to Mayer et al. (2019).

We further compute the same heat budget terms for each individual Arctic sub-region by considering the total advected heat as the sum of net OHT along each section line bounding the region. Given the full seasonal sea ice coverage (SIC > 90%) in the Laptev-East Siberian and Beaufort regions, we consider the net surface heat flux in each region as the sum of the ocean-ice heat flux (OIHF, $\mathrm{W}\,m^{-2}$) for the ice-covered fraction and SSHF ($\mathrm{W}\,m^{-2}$) for the remaining ice-free fraction of each grid cell. The regional heat budget analysis is computed from C-GLORSv5 data only, as the ocean-ice heat flux is not available for download from the ORAS5 reanalysis.

## 2.3 Ocean heat content

We next estimate ocean heat content ($Q$, $\mathrm{J}\,m^{-2}$) by vertically integrating monthly potential temperature over two target depths: the seasonally-varying mixed layer depth ($Q_{ml}$), to account for the energy stored at the atmosphere-ice-ocean interface, and



the 0-300 m depth layer ($Q_{300}$), to capture the intermediate depth layer characterized by the greatest AW warming (e.g., Shu et al., 2022) (3).

$$Q = \rho_0 C_p \int_{-Z}^{\eta} (T - T_{ref})dz \qquad (3)$$

where $T_{ref}$ is the reference freezing temperature = -1.8°C (Mayer et al., 2019) and Z represents the maximum depth bound-
ary. In calculating $Q_{ml}$, Z corresponds to the time-varying MLD, which is defined with the threshold method where potential density exceeds the surface reference value by 0.01 $kg\,m^{-3}$ (Peralta-Ferriz and Woodgate, 2015). The density difference criterion and the 0.01 $kg\,m^{-3}$ threshold have been widely tested in the Arctic (e.g., Toole et al., 2010; Gimbert et al., 2012; Timmermans et al., 2012; Thomson and Fine, 2003). We do not find any substantial differences in the spatial distribution and seasonality of MLD when adopting a different density criterion (e.g., 0.03 $kg\,m^{-3}$; not shown). In calculating $Q_{300}$, we
consider Z to be fixed at 300 m, or the ocean bottom if the water column is shallower than 300 m.

Finally, we obtain monthly anomaly time series of SIC, $Q_{ml}$ and $Q_{300}$ by removing a monthly climatological mean from each time series. To investigate how upper ocean heat content at both target depths influences sea ice variability on sub-seasonal time scale, lagged correlations are computed for each month and region, with the ocean leading sea ice by a lead time of one to three months. The time series used for the lagged correlation analysis are split between an earlier period (1980-2000) and a
later period (2001-2021) to assess the role of upper ocean warming and sea ice loss in recent decades. For comparison, we also show the lagged auto-correlation of SIC anomalies at each month as an indicator of the inherent sea ice predictability and how this varies across the two periods.

## 3  Results

### 3.1  Changes in the Arctic upper ocean between 1980-2021

Figure 2 shows annual mean trends in Arctic Ocean SIC, MLD, $Q_{ml}$, $Q_{300}$, SSHF and OIHF between 1980 and 2021 for the C-GLORSv5 reanalysis. Warming of the upper ocean is most evident in the Barents and Kara regions, both in terms of SIC reductions and positive trends in $Q_{ml}$ and $Q_{300}$, especially along the pathways of AW inflow (Fig. 2 a,c,d). The positive SSHF trend in the southern Barents Sea suggests reduced ocean cooling in this part of the region (Fig. 2e). Conversely, ocean heat loss to the atmosphere has been intensifying in the northern Barents and Kara seas and north of Svalbard along the pathway
of Fram Strait inflow. These changes suggest that the area of effective cooling in the Barents Sea has expanded towards the northern Barents and Kara seas (Shu et al., 2021). Significant reductions in OIHF are consistent with regions of sea ice loss along coastlines and in the southwest Barents Sea (Fig. 2f). The spatial distribution of MLD trends in the Barents and Kara seas (Fig. 2b) is consistent with that of SSHF. The overall weakening in stratification in the Atlantic Arctic sector is opposed to a modest decrease in MLD in the Canada basin, as corroborated by recent literature (e.g., Muilwijk et al., 2022).







**Figure 2.** Annual trends (1980-2021) in **(a)** sea ice concentration (SIC, %), **(b)** mixed layer depth (MLD, m) **(c)** ocean heat content in the mixed layer and in **(d)** the 0-300 m layer ($Q_{ml}$ and $Q_{300}$, $10^8$ J m$^{-2}$), **(e)** sea surface heat flux (SSHF, W m$^{-2}$) and **(f)** ocean-ice heat flux (OIHF, W m$^{-2}$) for the C-GLORSv5 ocean reanalysis. SSHF is defined as positive downward. Stippling denotes 95% significance.





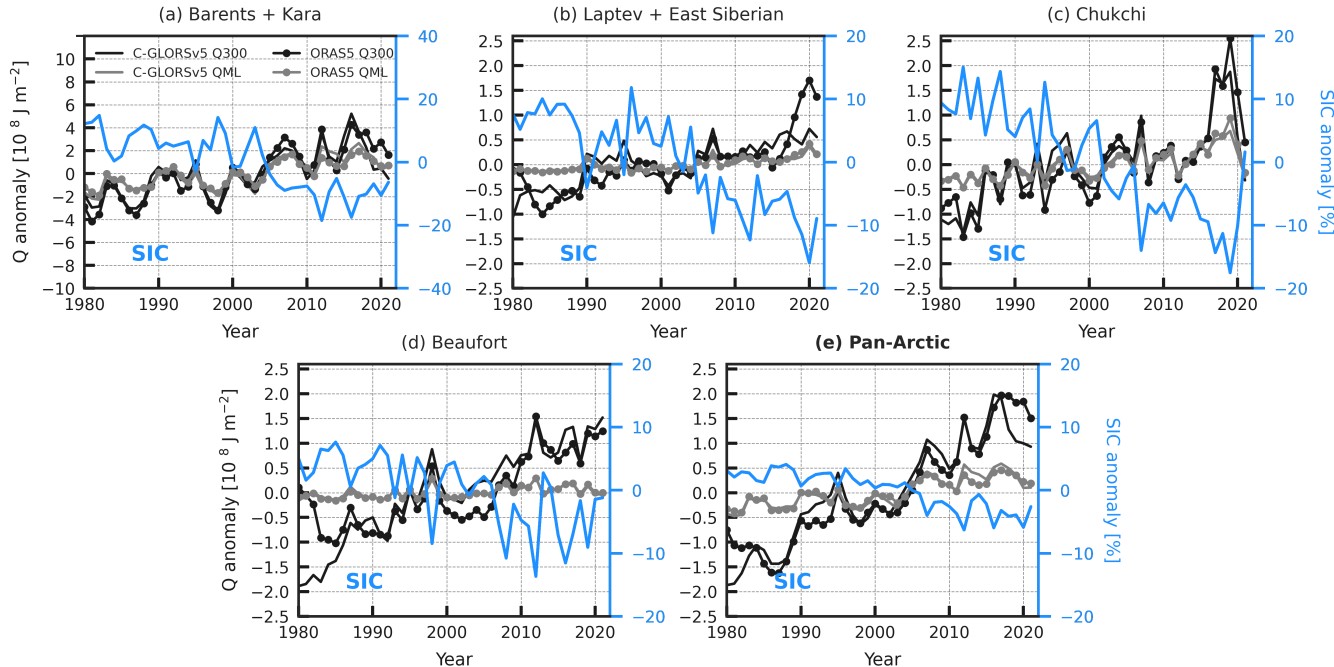

**Figure 3.** Anomaly time series of SIC, $Q_{300}$ and $Q_{ml}$ in C-GLORSv5 and ORAS5 for the Arctic regional seas and for the total Arctic. The pan-Arctic region **(e)** is defined as the area bounded by the Fram, BSO, Bering and Davis straits. Note the different y-axes used for the Barents-Kara region.

In Figure 3, annual anomaly time series of $Q_{ml}$, $Q_{300}$ and SIC for the four Arctic regional seas are presented. The variability of SIC and Q are intrinsically linked. A downward trend in SIC anomalies and an upward trend in $Q_{300}$ anomalies between 1980-2021 are evident for all regions as well as for the pan-Arctic mean (Fig. 3e), with a shift to negative SIC (positive $Q_{300}$) anomalies between the early and mid-2000s, depending on the region. The Beaufort Sea presents the weakest trend in SIC, presumably due to stronger upper ocean stratification. $Q_{ml}$ anomalies show an upward trend in the Barents-Kara, Chukchi and Pan-Arctic regions, while both the trend and variability of $Q_{ml}$ are comparatively small in the Laptev-East Siberian and Beaufort regions, as these regions are fully ice-covered for most of the year with limited surface mixing. It is worth mentioning the close agreement between the two reanalyses, with exception for a minor divergence in $Q_{300}$ at the beginning of the time series.

## 3.2 Assessment of budget closure

The net annual heat transport through the main Arctic gates is shown in Fig. 4 for C-GLORSv5 and ORAS5, and complemented in Table 2. The largest contribution to the total ocean heat transport into the Arctic comes from the BSO (62-99 TW C-GLORSv5; 57-96 ORAS5, values referring to annual means), followed by Fram Strait (14-47 TW C-GLORSv5; 19-38 TW



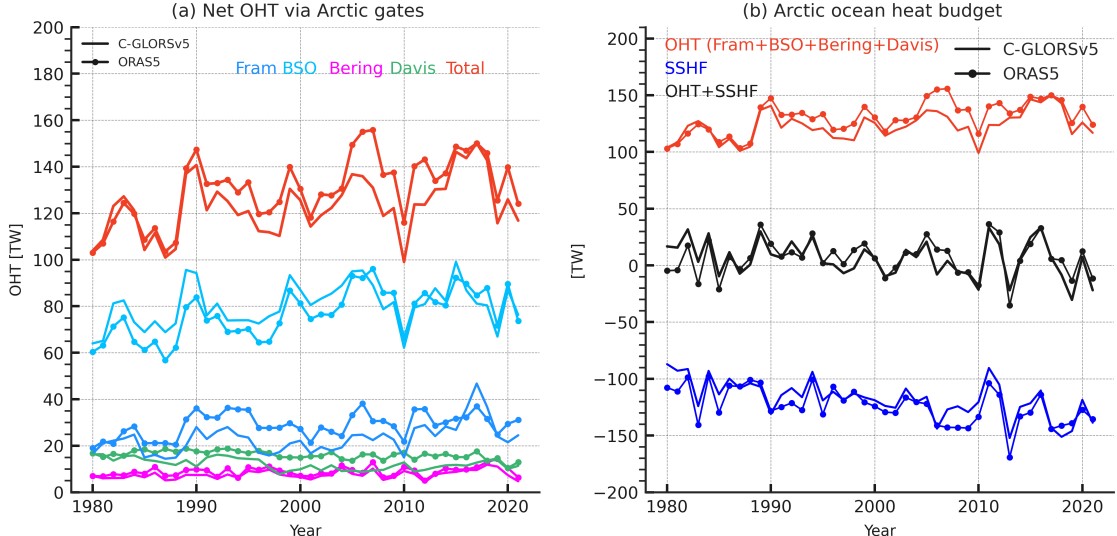

**Figure 4.** Annual time series of **(a)** ocean heat transport (OHT, TW) via the four main Arctic gates and their sum (red lines) and **(b)** pan-Arctic heat budget for C-GLORSv5 (solid) and ORAS5 (dotted). The locations of the Arctic gates are marked in Figure 1. Surface heat flux (SSHF) is weighted on the area bounded by the four gates and the negative sign indicates ocean heat loss. Black lines in panel **(b)** denote the sum of total ocean heat transport and surface heat flux.

ORAS5), Davis Strait (8-16 TW C-GLORSv5; 10-19 TW ORAS5) and Bering Strait (4-12 TW C-GLORSv5; 5-14 TW ORAS5). Transports along all sections show an upward trend for both reanalyses except for Davis Strait, where the net transport

has been decreasing (-0.29 GW $y^{-1}$ in C-GLORSv5 and -0.24 GW $y^{-1}$ in ORAS5). While there is a satisfactory agreement between the two reanalyses, C-GLORSv5 generally simulates smaller net transports via the Fram and Davis straits and a larger net transport via the BSO relative to ORAS5. As a result, the total Arctic OHT is slightly lower in C-GLORSv5 than in ORAS5 (99-150 TW and 103-155 TW, respectively).

We find OHT in C-GLORSv5 and ORAS5 to be consistent with the existing literature, though a direct comparison with

previous studies, especially observational ones, is challenging given the sensitivity of heat transport estimates to AW temperatures, data availability, and chosen reference temperature values. Our 1980-2000 estimates for Fram Strait and Davis Strait are in close agreement with the 1948-2007 multimodel mean of 22.9 TW and 13.4 TW, respectively, in CORE-II models (Ilıcak et al., 2016). We also find our BSO OHT estimate to be consistent with model results in Maslowski et al. (2004), which reported 74 TW in the same period (1979-2001) as well as with model simulations by Gammelsrød et al. (2009) and Sandø

et al. (2010).

Between 1980-2021, the total net OHT into the Arctic Ocean shows a positive trend in both reanalyses (1.21 GW $y^{-1}$ in C-GLORSv5 and 1.92 GW $y^{-1}$ in ORAS5). The increase in heat transport can also be appreciated by comparing monthly OHT estimates via each gateway averaged over the two halves of the timeseries (1980-2000 and 2001-2021; Table 2). The upward



**Table 2.** Average Net Ocean Heat Transport via Arctic Gates [TW][1]

| Source | Section/Gate | Period | Full depth | Mixed layer | 0-300 m |
|---|---|---|---|---|---|
| C-GLORSv5 | Fram Strait | 1980-2000 | 20.7 | 9.1 | 12.2 |
| | BSO | 1980-2000 | 77.4 | 37 | 72.9 |
| | Bering Strait | 1980-2000 | 7.1 | 3.1 | / |
| | Davis Strait | 1980-2000 | 13.4 | 3 | / |
| C-GLORSv5 | Fram Strait | 2001-2021 | 25.3 | 10.7 | 15.4 |
| | BSO | 2001-2021 | 83.2 | 39.1 | 78.2 |
| | Bering Strait | 2001-2021 | 8 | 3.5 | / |
| | Davis Strait | 2001-2021 | 10.9 | 2.6 | / |
| ORAS5 | Fram Strait | 1980-2000 | 27.6 | 14.5 | 16 |
| | BSO | 1980-2000 | 70.1 | 34.5 | 65.4 |
| | Bering Strait | 1980-2000 | 8.5 | 3.9 | / |
| | Davis Strait | 1980-2000 | 17.1 | 3.7 | / |
| ORAS5 | Fram Strait | 2001-2021 | 30 | 16.8 | 18.6 |
| | BSO | 2001-2021 | 83.1 | 35.7 | 78.2 |
| | Bering Strait | 2001-2021 | 9.4 | 4.2 | / |
| | Davis Strait | 2001-2021 | 15.1 | 3.2 | / |
| Observations | | | | | |
| Schauer et al. 2004 | Fram Strait | 1997–1999 | 16–41 | / | / |
| Skagseth et al. 2008 | BSO | 1997–2007 | 49.7 | / | / |
| Woodgate et al. 2010 | Bering Strait | 1991-2007 | 10-20 | / | / |
| Cuny et al. 2005 | Davis Strait | 1987-1990 | 17 | / | / |
| Model Simulations | | | | | |
| Lique and Steele, 2013 | Fram Strait | 1968-2007 | 20.9 | / | / |
| Maslowski et al. 2004 | BSO | 1979–2001 | 74 | / | / |
| Zhang et al. 2020 | Bering Strait | 1968–2009 | 9.7 | / | / |
| Muilwijk et al. 2018 | Davis Strait | 1890-2009 | 10 | / | / |

[1] Averages for C-GLORSv5 and ORAS5 are computed from monthly net heat transports. OHT for the 0-300 m layer is marked as '/' if the section depth is shallower than 300 m and values therefore correspond to the full depth OHT.





trend in OHT is offset by an overall decline in surface heat fluxes averaged over the pan-Arctic region (SSHF; -2.3 GW $y^{-1}$

in C-GLORSv5; -1.95 GW $y^{-1}$ in ORAS5), which is associated with enhanced ocean heat release to the atmosphere owing

to sea ice loss (Smedsrud et al., 2022). With the opposing trends of OHT and SSHF, the surface heat budget is closed around

zero over the historical time series (Fig. 4b), with good agreement between the two reanalyses. This ensures the satisfaction of

physical constraints and the realistic representation of heat fluxes in our data.

Annual time series of regional heat budgets (Fig. 5) further demonstrate the realistic representation of ocean heat advection

and surface heat fluxes in the C-GLORSv5 reanalysis. Net heat transport into the Barents-Kara region has been increasing at a

rate of 0.93 GW $y^{-1}$, chiefly due to the upward trend in BSO inflow. The increase in OHT is offset by a weak decline in surface

heat fluxes (SSHF+OIHF, -0.2 GW $y^{-1}$), which is however non-significant and levels off after 2010, indicating a reduction in

the overall cooling efficiency of the Barents-Kara region relative to the previous decades. Similarly, in all other regions, the

sum of advective and surface heat flux terms after the late 2000s marks a shift towards warmer conditions due to reduced ocean

heat loss. We find this to be largely driven by a sharp drop in OIHF in all regions, which is mostly associated with a decline in

sea ice concentration (Fig. 5e-h).

### 3.3    Ocean heat content as a precursor of SIC variability

In this section, we explore the role of upper ocean heat content as a precursor of SIC anomalies in the Arctic's regional seas

and how this has changed between the first (1980-2000) and second half (2001-2021) of our time series.

Figure 6 shows lagged correlations between SIC anomalies at each month and ocean heat content at both target depths ($Q_{300}$

and $Q_{ml}$) in the previous month, for the C-GLORSv5 reanalysis. The equivalent figure for ORAS5 is presented in Fig. A1 in

the Appendix. The bottom row of each plot shows the corresponding 1-month lagged auto-correlation of SIC anomalies. All

values shown in color are significant at the 95% level. We focus on the Barents-Kara and Laptev-East Siberian seas as these

regions have undergone the most remarkable changes in the Q-SIC interaction between the two periods. Regional differences

in the seasonal patterns of correlations are largely consistent with the seasonal cycle of sea ice in each region (dashed green

lines in Fig. 6). In the Barents-Kara region, SIC variability is closely linked to $Q_{300}$ and the seasonal peak of correlations

occurs in the freezing season, when AW inflow is at its annual maximum.

During the second half of the time series (2001-2021, Fig. 6e-h), changes in ocean warming and melt season duration are

reflected on SIC predictability. The 1-month lagged Q-SIC correlation is stronger and shifted towards the autumn, particularly

in the Laptev-East Siberian region. Relative to 1980-2000, the influence of $Q_{300}$ on SIC variability in the Barents-Kara region

during winter (Nov-Jan) 2001-2021 is considerably larger: for instance, the correlation of January SIC to December $Q_{300}$

increased from $r$=-0.72 in 1980-2000 to $r$=-0.83 in 2001-2021 in C-GLORSv5 (from $r$=-0.73 to $r$=-0.86 in ORAS5), while the

December to January SIC auto-correlation is not significant in either period. Though with considerable regional differences,

correlations between sea ice and ocean heat content anomalies are also significant at 2 months lag (Fig. A2 and A3, which

show 2-month lagged Q-SIC correlations for C-GLORSv5 and ORAS5, respectively) and at 3 months lag (Fig. 9 and A4,

which show 3-month lagged Q-SIC correlations for C-GLORSv5 and ORAS5, respectively).



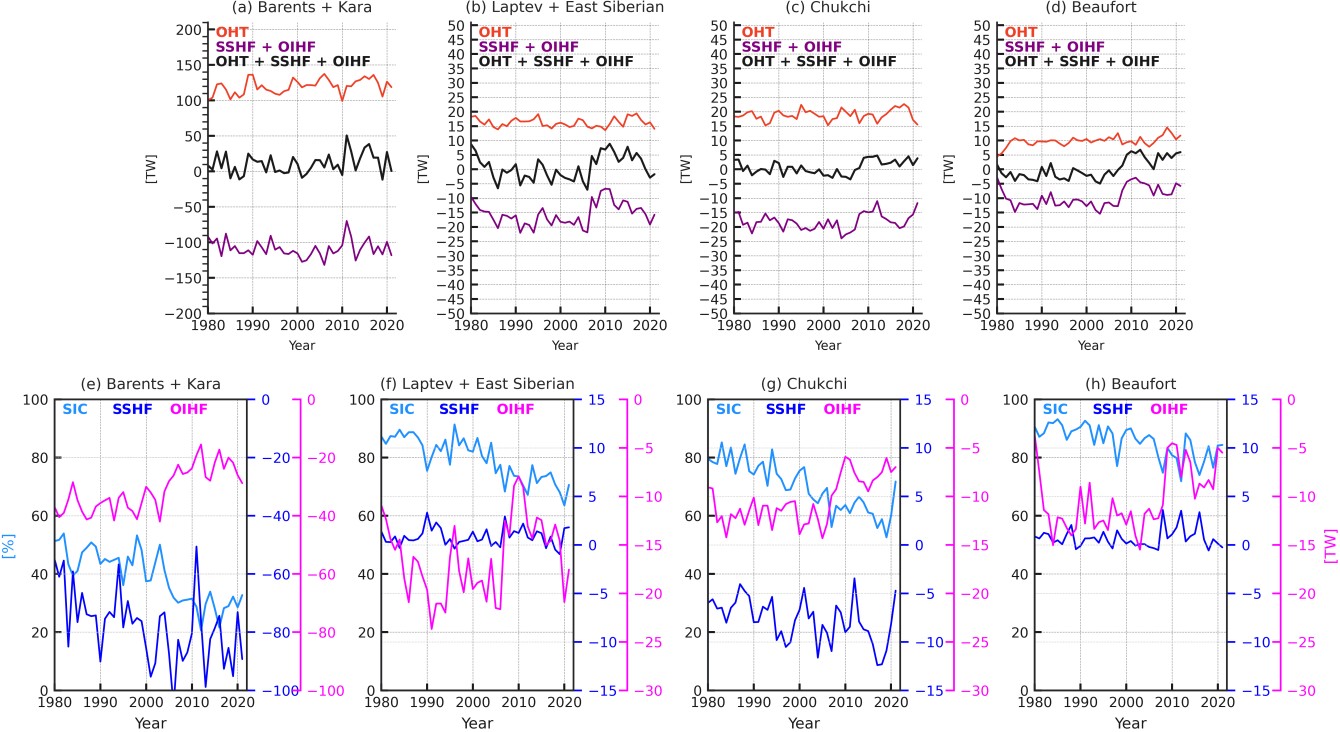

**Figure 5.** Time series of regional ocean heat budgets in the C-GLORSv5 reanalysis. **(a-d)** Annual time series of ocean heat transport, surface ocean heat fluxes and their sum. Ocean heat transport into a given region is defined as the sum of the net ocean heat transport along each section bounding the region, as marked in Figure 1. Surface heat fluxes (navy blue lines in **a-d**) are defined as the sum of net sea surface heat flux (SSHF) for the ice-free fraction of the grid cell and the ocean-ice heat flux (OIHF) for the ice-covered fraction of the grid cell. **(e-h)** shows the individual contribution of SSHF and OIHF in each region. Time series of sea ice concentration are shown in light blue. Note that here the sign of OIHF, which is positive from the ocean to sea ice, is reversed to match SSHF, indicating ocean heat loss.

Overall, the 1-month lagged SIC auto-correlation also increased in the second half of the time series, particularly in regions that have experienced the largest sea ice loss, i.e., the Barents-Kara region and Chukchi Sea (not shown). In the remainder of this section, we focus on the months where the 1-month lagged Q-SIC correlation is greater than the SIC auto-correlation, and thus where the predictive potential of ocean heat content is highest.

### 3.3.1 $Q_{ml}$ and autumn sea ice in the Pacific Arctic

The regional seas of the Pacific Arctic (Laptev-East Siberian, Chukchi and Beaufort seas) are characterized by complete or near-complete winter freeze-up and are considered to be areas of greater vulnerability to warming and lengthening of the melt season (Peng and Meier, 2018; Serreze et al., 2016). In these regions, the $Q_{ml}$-SIC anticorrelation maximizes in autumn (sea ice advance, Fig. 7) at 1-month lag time, however correlations are generally significant up to 3 months lag (not shown).





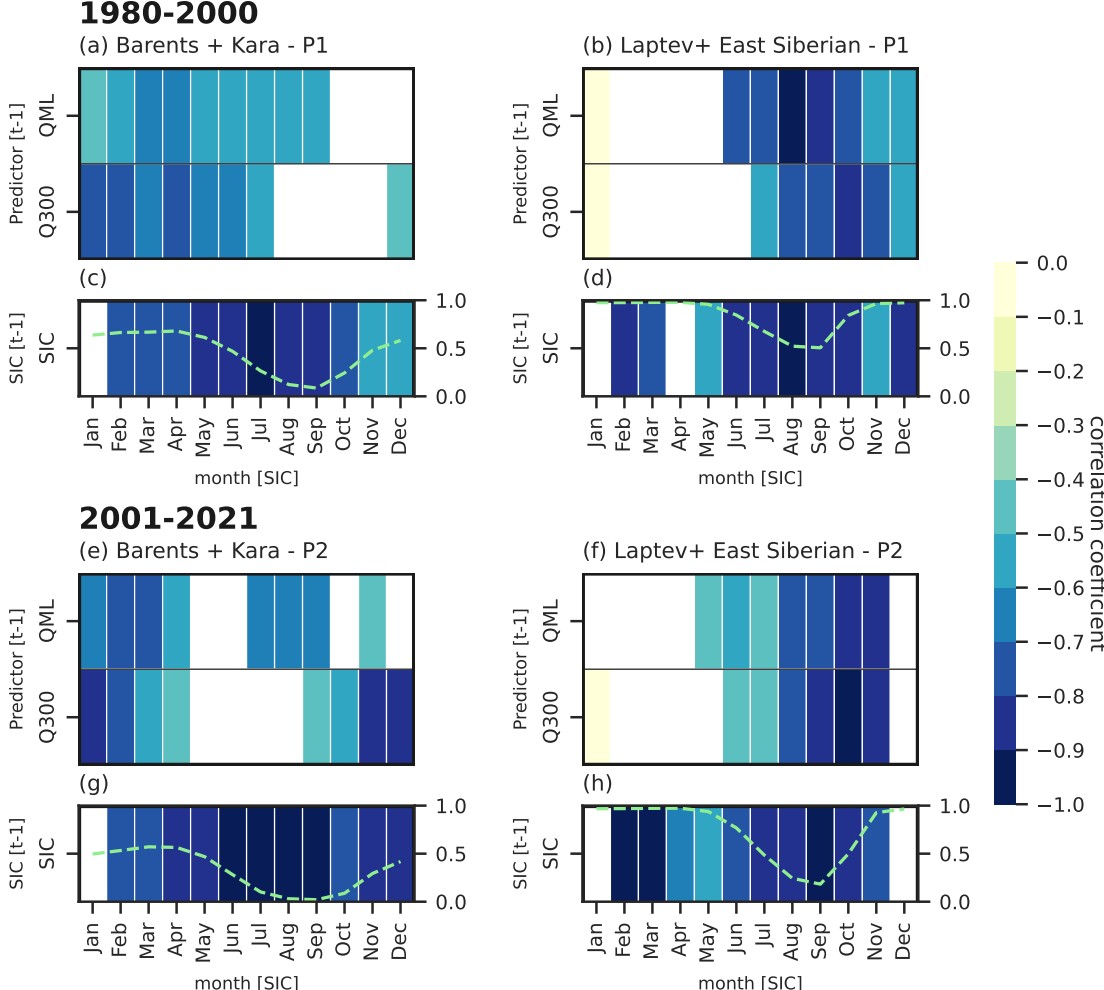

**Figure 6.** Lag 1 month correlations between SIC anomalies and ocean heat content anomalies in the 0-300 m layer and in the mixed layer, in the Barents-Kara and Laptev-East Siberian regions for the 1980-2000 **(a-b)** and 2001-2021 **(e-f)** periods, with the ocean leading sea ice. The 1-month lag SIC anomaly auto-correlation is shown in the bottom panels for comparison **(c-d, g-h)**. Green dashed lines indicate the SIC seasonal cycle in each region and period. Note that the sign of SIC auto-correlation is inverted. All correlations are significant at the 95% level. For brevity, only C-GLORSv5 is shown. The equivalent figure for ORAS5 is provided in the Appendix.

Figure 7 shows spatial correlation maps of regionally-averaged $Q_{ml}$ in October and the grid-point SIC anomaly in November in both reanalyses for 1980-2000 (Fig. 7 a,b) and 2001-2021 (Fig. 7 c,d). In all three regions, the anticorrelation between October $Q_{ml}$ and November SIC is higher than the October to November SIC auto-correlation. While this is true for both periods, there is a noticeable increase in the strength of the correlation in the Laptev-East Siberian region during 2001-2021, marking 250    a shift in the ice-ocean coupling from summer to autumn. Interestingly, the spatial pattern of correlation in the northeastern



Beaufort Sea indicates a reduction of the area of SIC prediction skill associated with $Q_{ml}$ during 2001-2021 (Fig. 7c,d). This area corresponds to the region of MLD shoaling during 1980-2021 (Fig. 2b).

The negative co-variability of Q and SIC anomalies in the shelf seas of the Pacific Arctic at the time of ice formation is consistent with the influence of the relatively warm summer Pacific water inflow through the shallow (~50m) and narrow (~ 85 km) Bering Strait (Rudels, 2015; Koenigk and Brodeau, 2014). However, we find greater difference in the predictability of November SIC between the first and second period in the Laptev-East Siberian region (from $r$=-0.52 to $r$=-0.86 in C-GLORSv5; from $r$=-0.51 to $r$=-0.78 in ORAS5) than in regions of greater PW influence, i.e., the Chukchi Sea (no change in C-GLORsv5; from $r$=-0.87 to $r$=-0.88 in ORAS5). This is consistent with the modest increase in Bering Strait inflow from 1980-2000 to 2001-2021 (Table 2) and the non-significant OHT trend in the Chukchi Sea (Fig. 5c). Furthermore, the time evolution of heat budget components in the Laptev-East Siberian region would suggest that the shift to warmer conditions after the mid-2000s is due to a decrease in ocean-ice fluxes associated with sea ice loss rather than an increase in OHT (Fig. 5f).

### 3.3.2 $Q_{300}$ and winter sea ice in the Barents-Kara region

In the Barents-Kara region, SIC and Q exhibit the highest negative correlation in the winter months, in agreement with existing evidence of the link between AW inflow and winter sea ice variability Årthun et al. (2019). December $Q_{ml}$ and $Q_{300}$ anomalies are both strongly anticorrelated with January SIC anomalies, although the correlation is higher for $Q_{300}$ (Fig. 8). An intensification of the $Q_{300}$-SIC link emerges in the 2001-2021 period (Fig. 8c,d), with correlations extending further east into the Kara Sea and over large part of the marginal ice zone in proximity of the St. Anna Trough. During 2001-2021, the southern Barents Sea is fully ice-free as the sea ice edge retreats north of 80°N and Atlantic water entering through the BSO is advected a longer distance before encountering sea ice. This change is consistent with the notion that a warming trend in AW inflow leads to less sea ice formation in the cold season (e.g., Long and Perrie, 2017), as previously demonstrated by the link between BSO heat transport and sea ice area ($r$=-0.76, Shu et al. 2021; $r$=-0.8, Li et al. 2017).

The predictability of regionally-averaged winter (Nov-Jan) SIC associated with $Q_{300}$ anomalies in the Barents-Kara region is maintained above $r$=-0.8 up to a lead time of 3 months in both reanalyses (Fig. 9a, A4a), highlighting the importance of ocean heat content as a precursor of SIC variability in this region, especially in months with inherently lower sea ice predictability.

## 4 Discussion and conclusions

The analysis presented in this study aims to shed light on the mechanisms underlying the regional patterns of upper ocean warming an its implications for sea ice variability. We used two eddy-permitting global ocean reanalyses, C-GLORSv5 and ORAS5, to investigate changes in ocean heat transport and surface heat fluxes through a heat budget analysis of the Arctic Ocean and its regional seas. Secondly, we assessed the role of Q anomalies in preconditioning sea ice variability by means of lag correlations. Results showed that while the total OHT into the Arctic has increased substantially between 1980-2021 (1.21 GW $y^{-1}$ in C-GLORSv5 and 1.92 GW $y^{-1}$ in ORAS5), only the Barents-Kara region is affected by a significant positive trend in ocean heat advection (0.93 GW $y^{-1}$), originating from the BSO. In this region, reduced ocean heat loss to the atmosphere




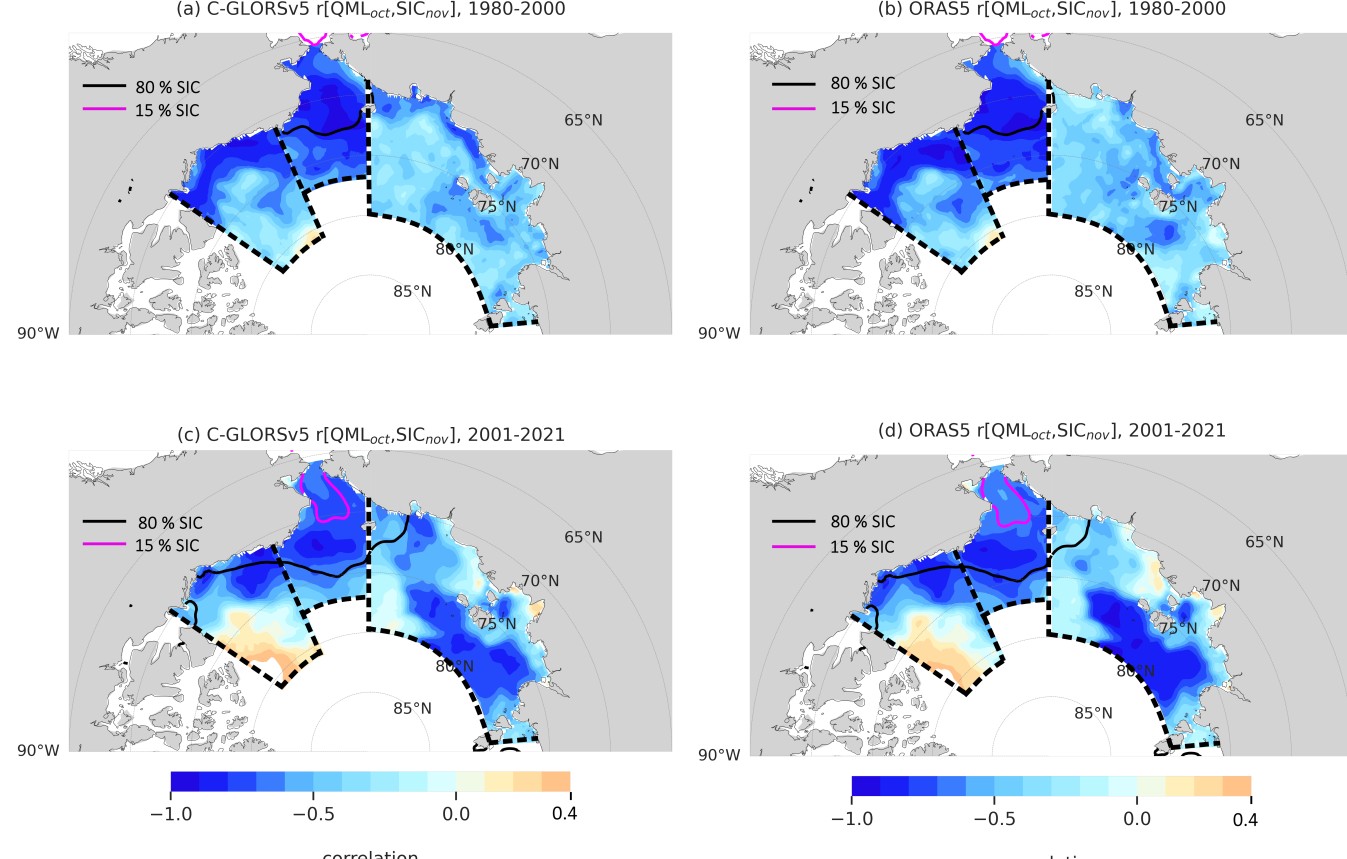

**Figure 7.** Correlation between the region-averaged $Q_{ml}$ in October and the grid-point SIC in November for the Laptev-East Siberian, Chukchi and Beaufort regions in 1980-2000 **(a,b)** and 2001-2021 **(c,d)** for C-GLORSv5 **(a,c)** and ORAS5 **(b,d)**. Black and magenta contour lines indicate the November climatological pack ice extension (80% SIC) and sea ice edge (15% SIC), respectively, over each period.

after the mid-2000s also contributed to the warming trend (Fig. 5a,e), in agreement with what found by Asbjørnsen et al. (2020). We have shown that Q anomalies in the Barents-Kara seas, particularly in the 0-300 m layer, act as an important precursor of

285    wintertime sea ice variability on sub-seasonal timescales. Relative to 1980-2000, this link intensified and expanded northwards and eastward during 2001-2021 (Fig. 7c,d), with close agreement between the two reanalyses (r[$Q_{300\,DEC}$,$SIC_{JAN}$]=-0.83 in C-GLORSv5 and -0.86 in ORAS5).

    In light of these findings, it becomes apparent that the evolution of sea ice variability in the Atlantic sector will depend on the opposing contribution of poleward heat advection and surface heat loss; in other words, whether the 'Barents Sea cooling

290    machine' will gradually lose its efficiency (Skagseth et al., 2020) or rather expand it to accommodate changes in AW inflow. For instance, CMIP6 simulations by Shu et al. (2021) showed that under the RCP8.5 scenario AW warming will increase and





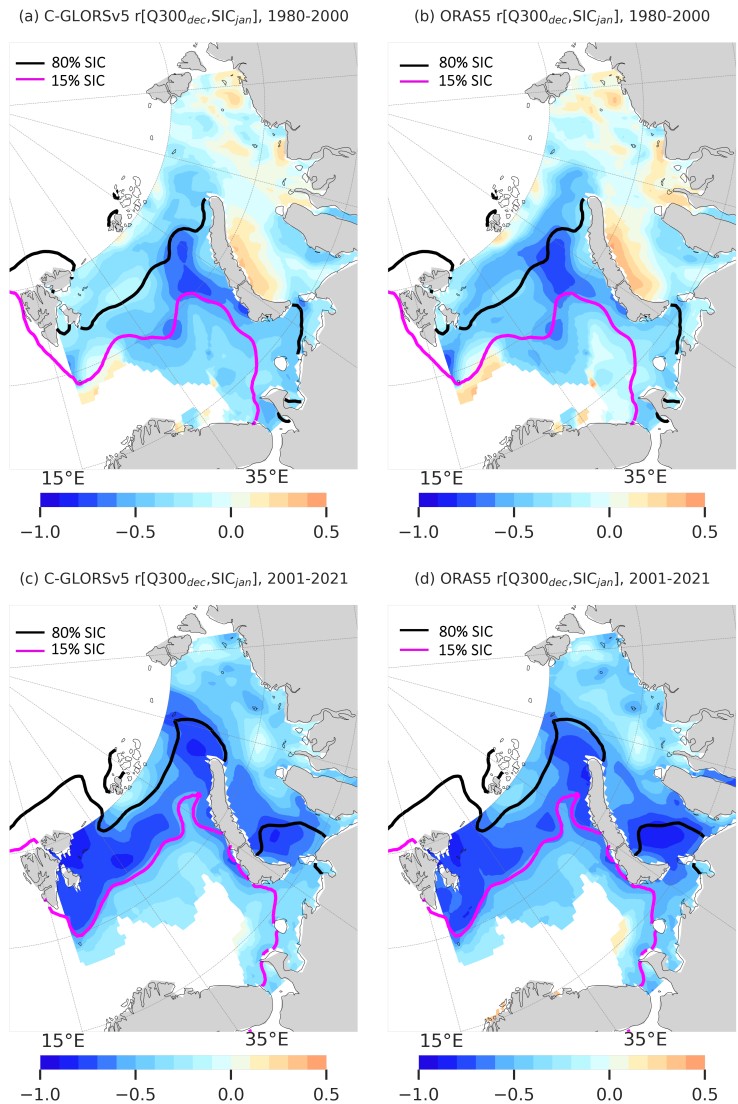

**Figure 8.** As in Figure 7, but showing the correlation between the region-averaged $Q_{300}$ in December and the grid-point SIC in January for the Barents-Kara region in 1980-2000 **(a,b)** and 2001-2021 **(c,d)** for C-GLORSv5 **(a,c)** and ORAS5 **(b,d)**. Black and magenta contour lines indicate the January climatological pack ice extension (80% SIC) and sea ice edge (15% SIC), respectively, over each period.

trends in winter surface heat loss and sea ice concentration will expand poleward, together with mixed layer deepening in the northern Barents Sea and and Kara Sea. According to this scenario, the $Q_{300}$-SIC coupling in the Kara Sea that emerged from our analysis between 2001-2021 (r < -0.5 in the western Kara Sea; Fig. 8) will likely intensify and expand northward and



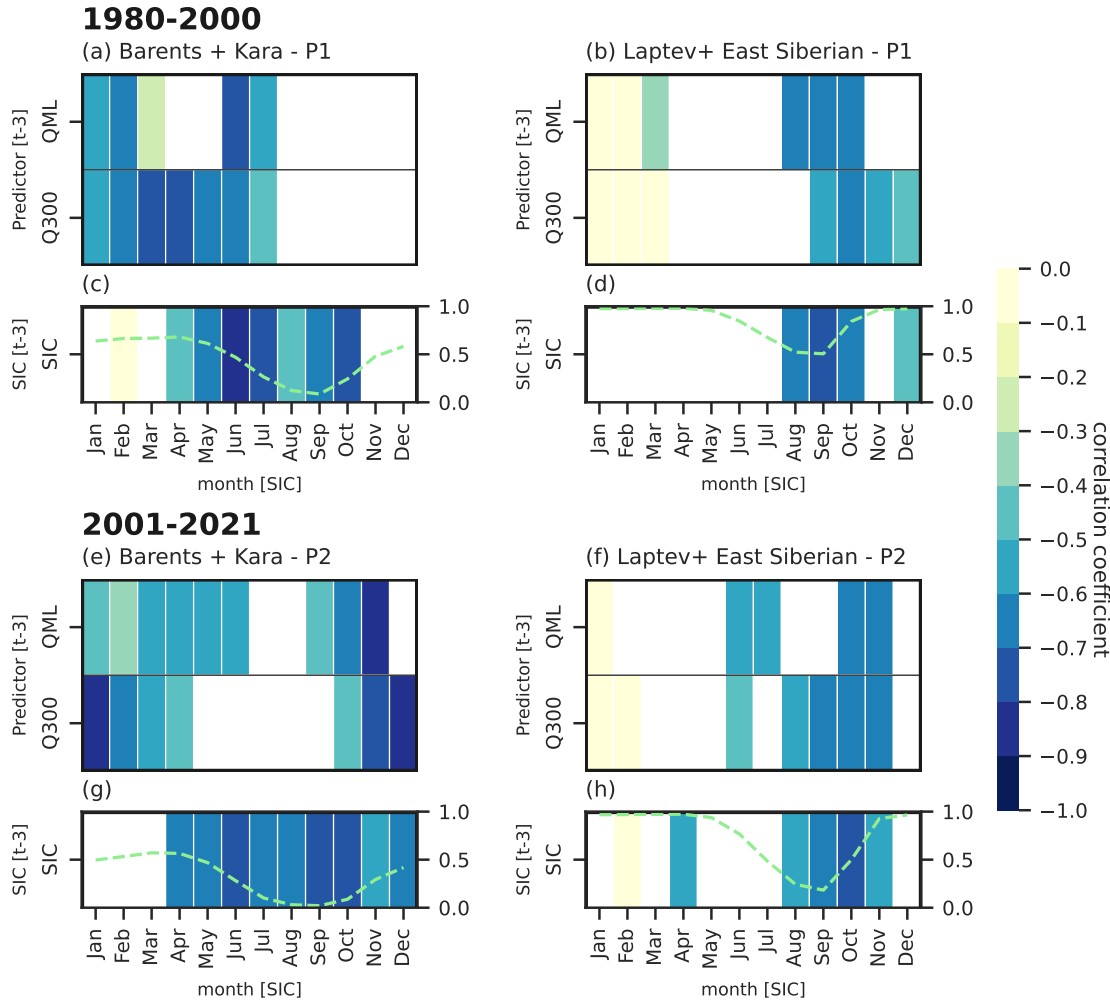

**Figure 9.** As in Figure 6, but for a lag time of 3 months, with the ocean leading sea ice.

eastward. This will likely allow for more local heat loss (negative SSHF trend) and enhanced convection as larger areas of open ocean become exposed to heat exchange with the atmosphere. According to Smedsrud et al. (2022), more heat loss will in turn accommodate for larger poleward AW inflow. However, substantial uncertainty remains around future projections of Arctic OHT, partly due to biases in AW representation in coupled climate models (e.g., Heuzé et al., 2023)

In the Laptev-East Siberian, Chukchi and Beaufort seas, we found no evidence of OHT having a substantial effect on upper ocean warming (i.e., OHT trends between 1980-2021 are not significant). Regional heat budgets point to a larger contribution of SSHF and OIHF to the overall warming, especially since the mid-2000s (Fig. 5). The variability of SIC during the warmer





months is strongly linked to that of Q at 1-3 lead months, particularly $Q_{ml}$. During 2001-2021, the timing of maximum correlation appears shifted from summer to autumn and the co-variability is heightened in the Laptev-East Siberian (notably, r[$Q_{mlOCT}$, $SIC_{NOV}$] in this region increased from -0.52 to -0.86 in C-GLORSv5 and from -0.51 to -0.78 in ORAS5). This finding is consistent with recent results by Sumata et al. (2023), who highlight a regime shift after 2007 consisting in a drastic reduction in September SIC in the area of sea ice formation of the Laptev-East Siberian region, with the consequent triggering of a widespread ice-albedo feedback. The timing of the regime shift agrees with the sharp drop in SIC anomaly in 2007 in the Laptev-East Siberian region (< -10%), concurrent with a shift to positive Q anomalies in both reanalyses (Fig. 3b). Hence, we suggest that enhanced summer absorption of atmospheric heat associated with sea ice loss and the resulting $Q_{ml}$ anomaly contributed to strengthening and delaying the peak of $Q_{ml}$-SIC anticorrelation during 2001-2021. While it is possible that an increase in Bering Strait inflow and the expansion of Fram Strait influence along the Siberian shelf may have additionally contributed to the stronger coupling, we found no evidence that this is the case.

It is worth noting that despite the long temporal coverage of C-GLORSv5 and ORAS5, the time series analyzed in this study are insufficient to infer patterns of decadal and multi-decadal variability in OHT and SSHF. It has been suggested that the recent increase in AW inflow could be in part associated with multidecadal fluctuations (e.g., Smedsrud et al., 2013), as earlier studies showed a similar warming during the 1930s-1940s (ETCW), followed by a period of relative cooling (Polyakov et al., 2004, 2005; Muilwijk et al., 2018). Internal climate variability indeed remains a source of substantial uncertainty for the long-term evolution of sea ice.

Additionally, the limited availability of observational datasets to be used as a benchmark implies that caution should be taken when interpreting the variability and trends of reanalysis data, especially when it comes to ocean-ice heat fluxes, which here could not be compared against other sources. Given the current trends in upper ocean warming, further work is necessary to better quantify the processes underlying the sea ice response to ocean heat surplus, particularly in regions where sea ice is transitioning to a state of higher vulnerability, i.e. the Kara and Chukchi Sea (Bliss et al., 2019). For instance, changes in the strength of stratification and halocline stability, which this study did not address, remain a large source of uncertainty due to substantial regional variability and model spread (e.g., Muilwijk et al., 2022; Pan et al., 2023; Shu et al., 2022).

Despite the aforementioned uncertainties, our results provide clear evidence that recent changes to the Arctic Ocean's surface heat budget have induced a strengthening of upper ocean-sea ice interactions in the Arctic regional seas. The strong coupling between ocean heat content and sea ice anomalies that emerged from our analysis has important implications for the sub-seasonal predictability of sea ice, which is of practical value to local communities and stakeholder groups, including for the navigability of Arctic shipping routes such as the Northern Sea Route (e.g., D'Angelo et al., 2021). Given the ongoing transition of the Arctic system to thinner, younger and more mobile sea ice, it is reasonable to expect increased vulnerability to continued warming of the upper ocean. Among other applications, ocean reanalyses have been successfully employed for the initialization of sea ice and ocean components of seasonal retrospective forecasts (e.g., Johnson et al., 2019; McAdam et al., 2022). Hence, continued efforts in the representation of upper ocean variables though improved accuracy of ocean models, atmospheric forcing and data assimilation schemes, together with the current expansion of the observational network (e.g.,



Tsubouchi et al., 2012), are crucial to help us address open questions while ensuring consistent monitoring of the Arctic Ocean climate.

*Data availability.* All data analyzed in this study are freely available online. ECMWF ERA5 can be downloaded from the Copernicus Climate Change Service (C3S) Data Store at https://cds.climate.copernicus.eu/cdsapp!/dataset/reanalysis-era5-single-levels?tab=overview.

340    The CMCC C-GLORSv5 Reanalysis can be downloaded from the PANGAEA repository (http://c-glors.cmcc.it/index/index-9.html?sec=0). The ECMWF ORAS5 Reanalysis is available from C3S at https://cds.climate.copernicus.eu/cdsapp!/dataset/reanalysis-oras5?tab=form. Both C-GLORSv5 and ORAS5 reanalyses can also be found on the Copernicus Marine Service website (CMEMS) as part of the Global Ocean Ensemble Physics Reanalysis (Product Identifier: GLOBAL REANALYSIS PHY 001 031).



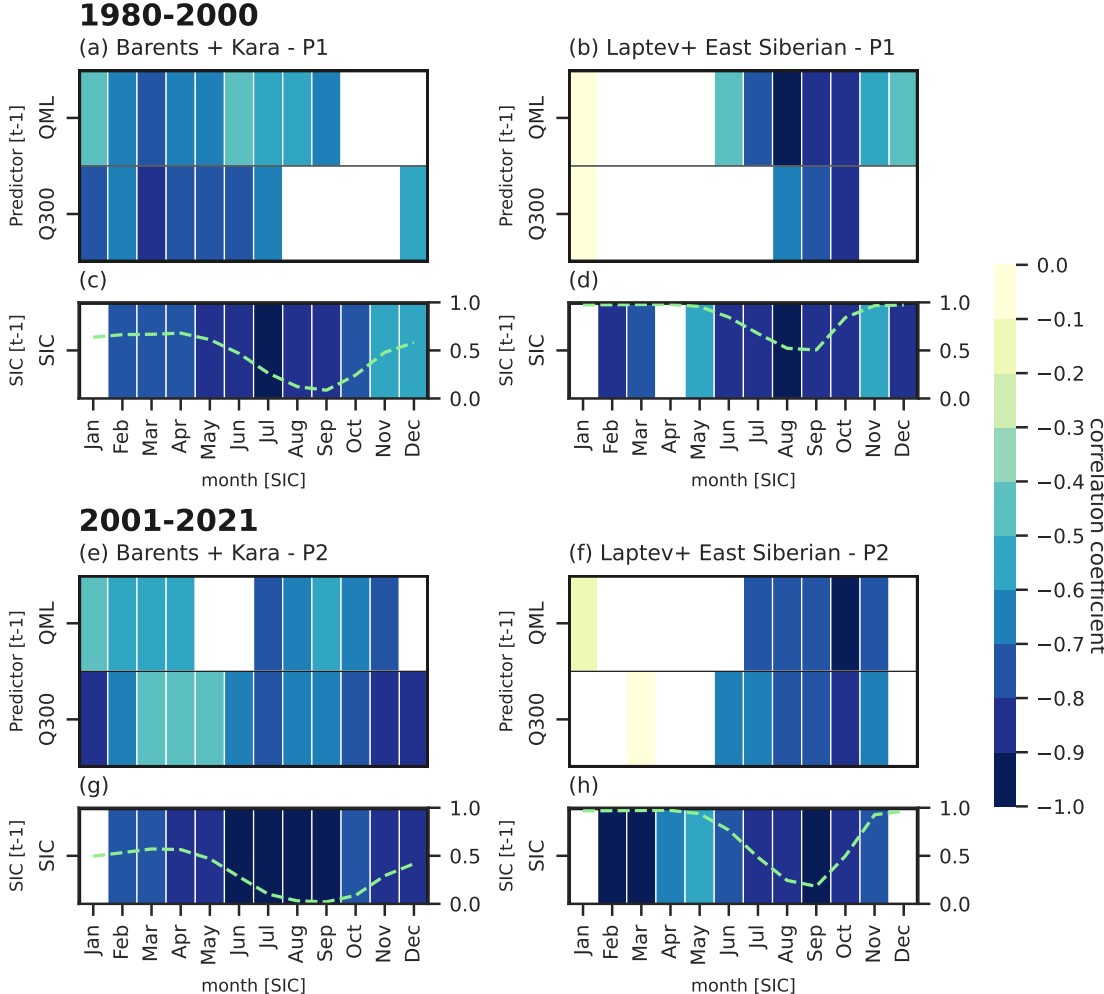

**Figure A1.** Lag 1 month correlations between SIC anomalies and ocean heat content anomalies in the 0-300 m layer and in the mixed layer, in the Barents-Kara and Laptev-East Siberian regions for the 1980-2000 (a-b) and 2001-2021 (e-f) periods, with the ocean leading sea ice for the ORAS5 reanalysis. The 1-month lag SIC anomaly auto-correlation is shown in the bottom panels for comparison (c-d, g-h). Green dashed lines indicate the SIC seasonal cycle in each region and period. Note that the sign of SIC auto-correlation is inverted. All correlations are significant at the 95% level.



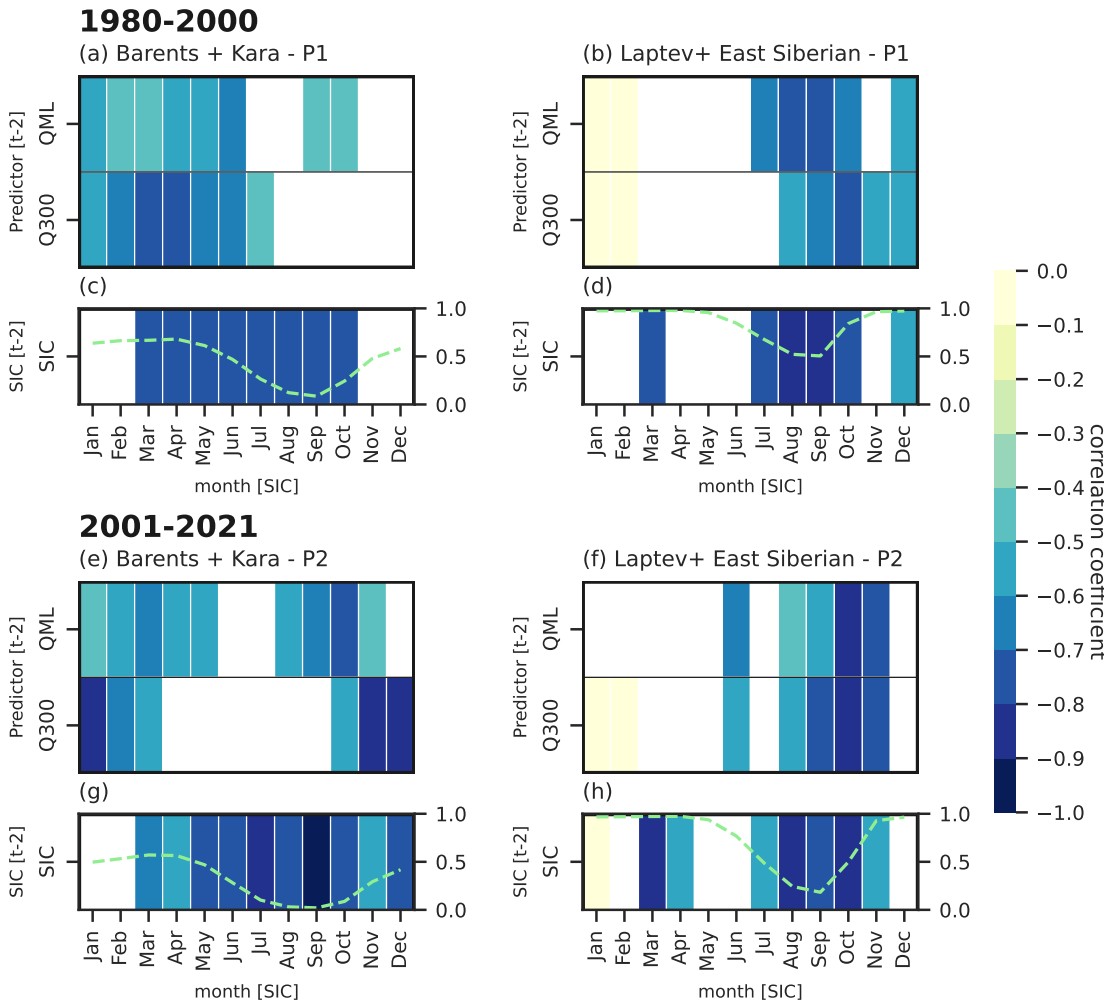

**Figure A2.** Lag 2 month correlations between SIC anomalies and ocean heat content anomalies in the 0-300 m layer and in the mixed layer, in the Barents-Kara and Laptev-East Siberian regions for the 1980-2000 (a-b) and 2001-2021 (e-f) periods, with the ocean leading sea ice for the C-GLORSv5 reanalysis. The 2-month lag SIC anomaly auto-correlation is shown in the bottom panels for comparison (c-d, g-h). Green dashed lines indicate the SIC seasonal cycle in each region and period. Note that the sign of SIC auto-correlation is inverted. All correlations are significant at the 95% level.





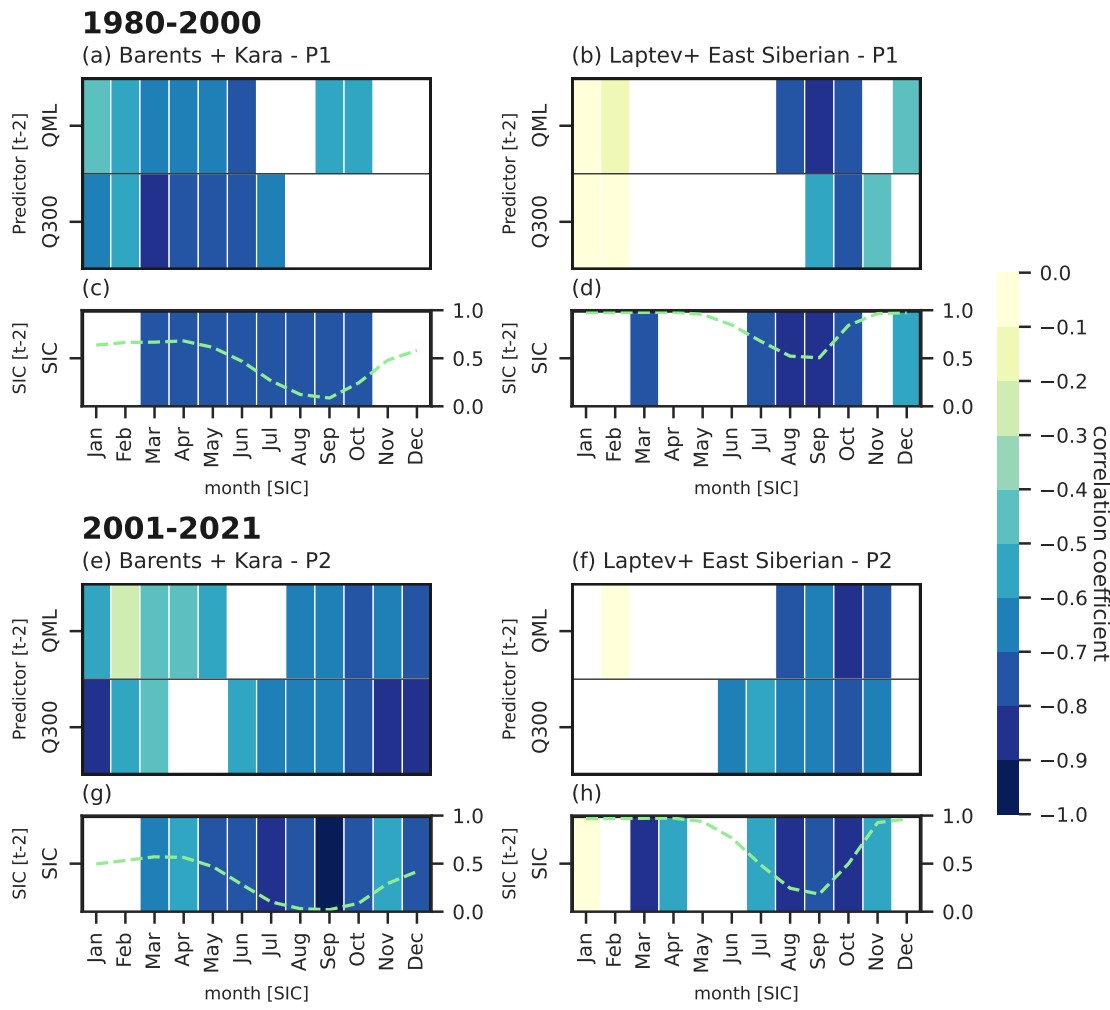

**Figure A3.** As in A1, but for lag 2 months.



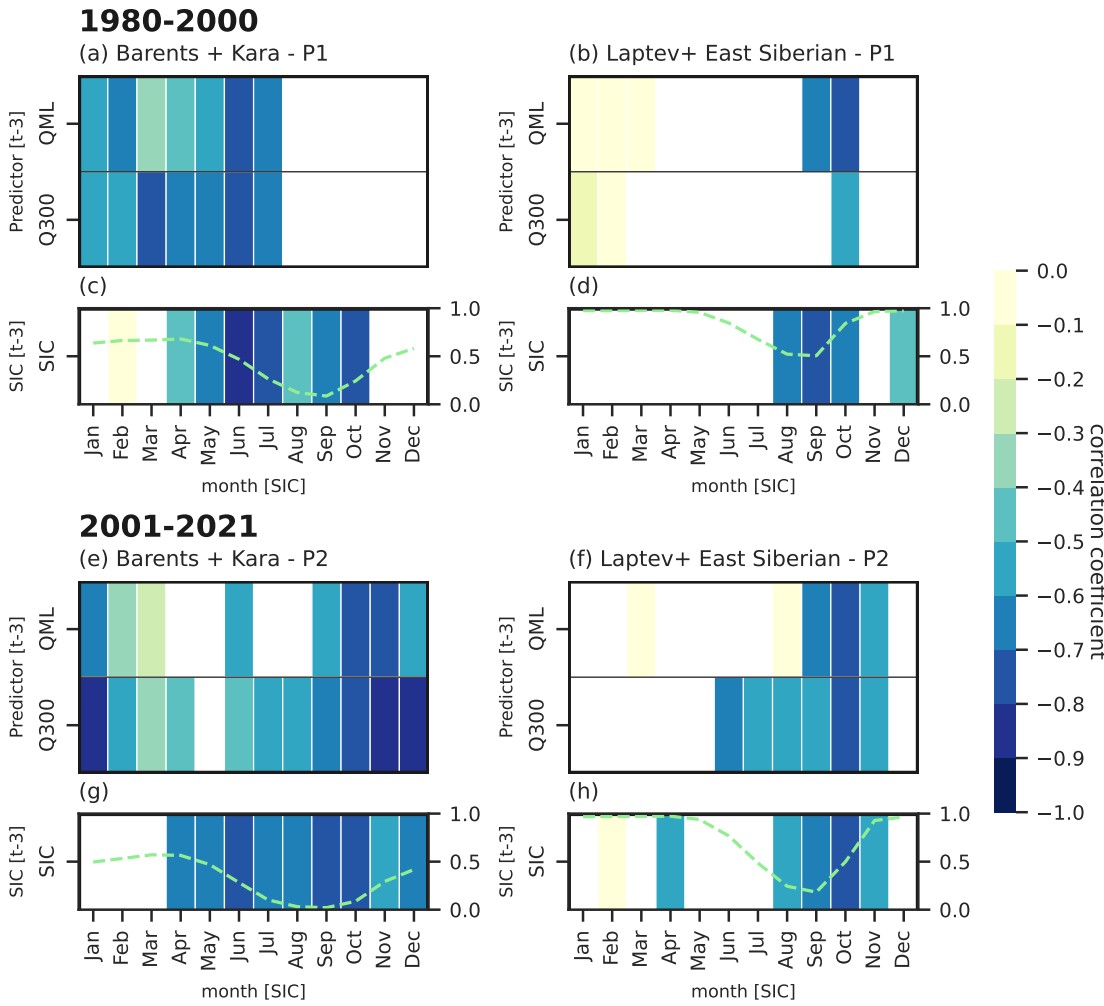

**Figure A4.** As in A1 and A3, but for lag 3 months.



345 *Author contributions.* EB processed and analyzed the data, made the figures and wrote the manuscript. DI conceived the study and contributed to the interpretation of findings. SMas, SMat and PR equally contributed to interpreting, discussing and improving the results of the study. All authors reviewed the manuscript.

*Competing interests.* The authors declare that they have no conflict of interest.

*Acknowledgements.* We thank Ronan McAdam and Deep S. Banerjee for providing valuable insight into the calculation of ocean heat content
350 and transport in the C-GLORSv5 and ORAS5 reanalyses.



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
