# Peer review of "The role of upper ocean heat content in the regional variability of Arctic sea ice at sub-seasonal time scales"

_EGUsphere, 2023_

## Author Response (AR1)

**Response to Anonymous Reviewer 1**

**General Response:**

We thank the reviewer for providing helpful suggestions on how to improve the manuscript. We have addressed the general and specific points raised in the review by editing part of the text to clarify some statements and by improving some of the figures.

We hope our edits to the manuscript are satisfactory and our responses are thorough. What follows is a response to the general points raised by the reviewer with reference to how they have been addressed. Point-by-point responses to specific comments are also included below. Line and page numbers refer to the version with track changes.

***Reviewer's comment**: This manuscript is written in good grammar and contains technically good quality figures. The analysis appears generally valid, and lagged correlation values greater than 0.7 (explaining more than 50% of the variance) indicate significant influences of Q on SIC at sub-seasonal scales, which is interesting. It is not clear what new the results contain in terms of mutual significance of physical processes identified, the study seems to confirm earlier results, and therefore the novelty aspect is low, although it claims to shed light on the regional mechanisms (line 276). This aspect could possibly be elevated by adding more specific regional interpretations of the roles of physical processes, following the example presented in lines 309-312, but at a more advanced level. Particularly, the Arctic warming, linked to sea-ice variability, is a result on multiple other processes than the ice-albedo feedback, such as the atmospheric ones related to large-scale weather patterns, winds, precipitation and air temperature, and riverine heat influx. What is the role of processes other than OHT on sub-seasonal sea-ice variability?*

**Response**: We thank the referee for the comment. However, the main goal of our work is to advance our knowledge of how upper ocean heat content (Q) within the Arctic basin influences sea ice variability in the context of the general warming trend of the Arctic Ocean, and the implications this has on sea ice predictability. In our view, one of the advantages of using Q as a sub-seasonal predictor of SIC is that this quantity encompasses a multitude of interrelated processes affecting the Arctic Ocean, such as variations in stratification and salinity, freshwater fluxes, wind forcing and precipitation. Nevertheless, determining the specific physical mechanisms that drive upper ocean warming is beyond the scope of the study, and so is investigating all sub-seasonal drivers of sea ice variability. The relative influences of the aforementioned parameters (rivering influx, precipitation, wind forcing) vary greatly across regions and not all of them are relevant at sub-seasonal timescales. So, quantifying their roles would require a separate study (e.g., focusing on a single region). We appreciate this suggestion and hope to include these aspects in future work.

**Changes to the Manuscript**: To address the above comment, we clarified the scope of our study in the introduction section (L73-77, P.3) and added a more detailed discussion of the potential impact of additional mechanisms that are not explicitly addressed in our analysis (L332-339, P.19; L368-374, P.20).

***Reviewer's comment****: Additionally, now only the sea-ice variability in terms of SIC is explored, while the thinning of ice could be linked to SSHF.*

**Response**: Sea ice thickness and volume are certainly affected by upper ocean warming, and determining the degree of sea ice thinning associated with Qml anomalies is an interesting question to explore in future work. In our study, the choice of metric for the lag correlation analysis in Sec 3.3 is justified by the fact that SIC is the most well-constrained sea ice diagnostic (e.g., Ivanova et al. 2014). Satellite observations of SIC exist since 1979 and they are used to constrain SIC in ocean and atmospheric reanalyses, which substantially reduces the uncertainty around this variable. On the other hand, sea ice thickness measurements continue to be affected by the scarcity of observed data, which are not spatially and temporally uniform. Uncertainties in sea ice thickness among available ocean reanalyses remain large (e.g., Uotila et al. 2019). Among other examples, this emerges clearly from the ensemble spread of CMEMS global ocean reanalyses (GREP), which is much larger for SIT than SIA (Figure **R1** attached).

[Figure]

**R1**: Time series of Arctic SIA (upper panel) and SIT (lower panel) between 1993-2020 for different reanalysis products as part of the GREP ensemble.

Ivanova, N., Johannessen, O. M., Pedersen, L. T., & Tonboe, R. T. (2014). Retrieval of Arctic sea ice parameters by satellite passive microwave sensors: A comparison of eleven sea ice concentration algorithms. IEEE Transactions on Geoscience and Remote Sensing, 52(11), 7233-7246.

Uotila, P., Goosse, H., Haines, K., Chevallier, M., Barthélemy, A., Bricaud, C., ... & Zhang, Z. (2019). An assessment of ten ocean reanalyses in the polar regions. Climate Dynamics, 52, 1613-1650.

*Reviewer's comment: Also, sea-ice dynamics, not only thermodynamics, could be regionally important, as the Arctic sea-ice volume declines. In this context, adding a reanalysis product with sea-ice model having a sub-grid-scale sea-ice thickness distribution parametrisation would be valuable. A considerable issue is that the study is based on two very similar ocean reanalysis products only, both driven by ERA-Interim atmospheric forcing. But ocean reanalyses are known to vary a lot in the Arctic, see for example Uotila et al. (2019) Figure 8. Related to this, the ERA-Interim sea-ice parametrisation is not particularly realistic (Batrak et al. 2019), that affects its surface energy balance and further the variables used to drive ocean models. Other more contemporary atmospheric reanalyses and ocean reanalyses driven by them might do better, which would be interesting to know. Importantly, the robustness of the quantitative results remains questionable before including more reanalyses, or similar data, and therefore must be tested with other products, such as TOPAZ, NCEP CFSR, ECCO2, GLORYS, NEMO-EnkF and PIOMAS, before the publication. Many reanalyses have output publicly available. In particular, the NCEP CFSR based comparison would be valuable as it is a coupled atmosphere-ocean one, with the ocean able to influence the atmosphere. Findings from this extended analysis could then be compared to the ones by Mayer et al. (2019) in discussion.*

**Response**: Using a larger number of data products, provided that these meet the requirements for a meaningful comparison, is certainly an added value in this type of study and we appreciate the referee's suggestion. However, other available ocean reanalyses have the limitation of covering a shorter time period, and/or having lower vertical resolution, both of which were elements of priority in our choice of data products for this study. In particular, using data products with reduced temporal coverage (e.g., TOPAZ, GLORYS and ECCO2 provide data starting from the early 1990s; NCEP CFSR from 2011, for most variables) would prevent us from analyzing the change in Q-SIC coupling between the late 20th century (1980-2000) and the early 21st century (2001-2021), which is in our view one of the aspects with broader implications in this study. In addition to the extensive temporal coverage, C-GLORSv5 and ORAS5 have the advantage of being eddy-permitting ocean reanalyses with high vertical resolution (75 levels). In comparison, TOPAZ and ECCO2 extend for 50 vertical levels, whereas CFSR includes 64 levels in the atmosphere model and 40 in the ocean model. During the early stages of this work, we compared our results with data from the SODA3 ocean reanalysis (http://www.soda.umd.edu/), which provides data since the 1980s and has a vertical resolution of 50 levels. However, we realized that the vertical distribution of the ocean layers in SODA3 provided insufficient detail into the representation of the shallow mixed layer of the Eastern Arctic during summer (Figure **R2c**, showing an example of SODA3 summer MLD in the Arctic, compared to C-GLORSv5 and ORAS5), which created an issue in our calculation of Qml during summer (Figure **R2d**). For this reason, we restricted our study to high-resolution ocean reanalyses with sufficient detail in the surface layer, as we believe these are the best-suited products for this type of study.

**Changes to the Manuscript**: To address this point, we added a more detailed explanation for why we chose to use C-GLORSv5 and ORAS5 reanalyses in our study (L86-88, P.3).

[Figure]

**R2**: (a-c) Example of mixed layer depth (m) in August 1980-2021 in the C-GLORSv5, ORAS5 and SODA3 reanalyses. (d) example of Qml in June 1980 for the SODA3 reanalysis.

*Reviewer's comment: AW is mentioned many times as the source of OHT but apparently without consideration of its depth. In some parts of the Arctic Ocean AW is lies deeper under a strong halocline and its influence on upper ocean Q and SIC variability may remain limited. This should be taken into account when interpreting the results but has not been mentioned.*

**Response**: We thank the reviewer for the opportunity to clarify this aspect. OHT in our heat budget analysis (Section 3.2) is computed over the full water column, as this is the quantity required to close the heat budget. In Table 2 (Page 10), however, we included estimates of OHT for different depths (i.e., OHT in the mixed layer and OHT for the 0-300m layer). For the Atlantic straits (Fram and BSO), these values show that the largest amount of AW heat is transported in the 0-300m layer, i.e, there is little difference between the full depth OHT and the 0-300m OHT, especially in the BSO. Although the 0-300m layer can include the core of AW inflow in the Atlantic Arctic sector, it can be assumed that AW water can be deeper and that not all heat contained in the 0-300m layer (Q300) is always of AW origin. This is because the water column undergoes substantial mixing downstream and the portion of AW that directly interacts with the surface depends on stratification and halocline stability. However, our main goal is to investigate how Q300 influences sea ice at sub-seasonal timescales, somehow independently from its source. The strengthening of this coupling in the Atlantic Arctic sector (Section 3.3.2), concurrent with an upward trend in OHT in the Barents/Kara region, is interpreted in the context of a stronger AW influence.

**Specific points**

*Reviewer's comment*: Line 133. Is mentioned that the diffusive heat transport is not taken into account. Does it mean that AW is not diffusing heat to upper ocean, which could be a significant shortcoming?

**Response**: In the heat budget equation (Eq.1) Qdiff represents the lateral heat diffusion term, which can originate from lateral turbulent mixing and diapycnal mixing. As this quantity is generally considered to be negligible compared to the advective and surface heat flux terms (OHT and SSHF; e.g. Lique and Steele, 2013), we omit it in Eq.1. However, our analysis of the Q-SIC coupling (Section 3.3) is based on the vertical integration of C-GLORSv5 and ORAS5 potential temperature, a quantity that incorporates the exchange of all heat flux terms resolved by the NEMO ocean model.

Lique, C., & Steele, M. (2013). Seasonal to decadal variability of Arctic Ocean heat content: A model-based analysis and implications for autonomous observing systems. Journal of Geophysical Research: Oceans, 118(4), 1673-1695.

*Reviewer's comment*: Line 149. Did you consider setting Tf to depend on salinity? In the Arctic Ocean some regions have a rather low salinities and therefore low Tf and possibly significantly lower Q than with Tf=-1.8 degC.

**Response**: Since the quantities we consider for the analysis in Section 3.3 are only trends and anomalies, our results are not directly affected by the choice of reference temperature in Eq. 3. The freezing point in the sea ice model can depend on the surface ocean salinity and the sea ice salinity. In this ice code, the ice salinity is kept constant as the salt released to the ocean; the freezing point can depend on surface salinity via an empirical constant or be optionally fixed (in our case to -1.8°C). The choice of Tf = -1.8°C is motivated by the intent to conform with the existing literature (e.g., Mayer et al. 2019), where Q is typically computed by removing the constant of freezing temperature.

Mayer, M., Tietsche, S., Haimberger, L., Tsubouchi, T., Mayer, J., & Zuo, H. (2019). An improved estimate of the coupled Arctic energy budget. Journal of Climate, 32(22), 7915-7934.

*Reviewer's comment*: Line 168. Figure 2e and 2f show annual trends, but seasonal trends would be better to show, and in general seasonal summer/winter analyses where relevant instead of the annual one. This is because SSHF represents ocean cooling in winter but warming in summer. The annual analysis becomes especially problematic when showing flux anomalies/trends, where a negative (positive) anomaly/trend could be related to either a downward flux becoming stronger (weaker) downward, or an upward flux becoming less (more) upward. The same issue exists at least in Figure 3.

**Response**: We agree with the referee. We updated Figure 2 in the manuscript (Page 8), such that the 1980-2021 trend in SSHF is now split between melting and freezing seasons (MJJ-OND), to facilitate the interpretation of changes in the surface fluxes. OIHF on the other hand does not show as great a seasonal variation, as the trend is mostly associated with regions of reduced ice cover.

*Reviewer's comment*: Figure 2. Stippling denotes 95% significance, but what is confusing is that even zero trends are often stippled, for example in Figure 2c. Such significance testing is meaningless and should not be used.

**Response**: Thank you for pointing this out. This issue was fixed in the updated version of Figure 2.

*Reviewer's comment*: There are quite striking differences between C-GLORS and ORAS5 Q300 in Figure 3b and 3c, which has not been mentioned and reasons discussed. Also, instead of anomalies would be better to plot absolute Q values to reveal the real differences between the reanalyses. It

*would also be good to add the basin averaged sea-ice thickness time series to show its relationship to Q:s.*

**Response**: Following this suggestion, we replaced Figure 3 in the manuscript (P.11) with a version showing absolute values of SIC, Q300 and Qml, instead of anomalies. A version of Figure 3 including the annual pan-Arctic sea ice volume thickness obtained from PIOMAS data (Fig. **R3e**) is attached. In terms of differences between the two products, C-GLORSv5 and ORAS5 diverge slightly in the Laptev/East Siberian Q300 due to higher temperatures in C-GLORSv5 (the two products assimilate different observations, so minor differences are not surprising). However, both the trend and interannual variability of Q300 are very similar in the two products. To address this comment, we mentioned this discrepancy in the text in L204-205, P.9.

[Figure]

**R3**: *Revised version of Fig. 3 in the original manuscript, showing absolute values of SIC, Q300 and Qml in each Arctic sub-region. The green line in panel (e) shows pan-Arctic sea ice thickness (m) obtained from PIOMAS.*

**Reviewer's comment**: *Lines 207-208. The expression 'satisfaction of physical constraints' is unclear and should be clarified.*

**Response**: Following the suggestion, we modified this sentence in L232-234, P.12: "With the opposing trends of OHT and SSHF, the surface heat budget is closed around zero over the historical time series (Fig. 4b), with good agreement between the two reanalyses."

**Reviewer's comment**: *Lines 215-216. Links between the OIHF sharp drops and SIC decline are not clear in Figure 5. For example in Figure 5f the SIC decline starts years earlier than the OIHF sharp drop. So, the statement looks rather speculative.*

**Response**: This statement was reworded in L240-242, P.12: "Similarly, in all other regions, the sum of advective and surface heat flux terms after the late 2000s marks a shift towards warmer conditions due to reduced ocean heat loss. We find this to be largely driven by a sharp drop in OIHF in all regions, concurrent with a decline in sea ice concentration (Fig. 5e-h)"

*Reviewer's comment*: Lines 220-223. This text repeats Figure 6 caption and should be removed.

**Response**: Addressed in L247-249, P.13: "Figure 6 shows 1-month lag correlations between SIC anomalies at each month and ocean heat content at both target depths (Q300 and Qml), along with 1-month lag SIC anomaly auto-correlations, for the C-GLORSv5 reanalysis. The equivalent figure for ORAS5 is presented in Fig. A1 in the Appendix."

*Reviewer's comment*: Line 227. Specify the freezing season months. Mentioning the AW annual maximum indicates the AW seasonal variability, add a suitable literature reference here.

**Response**: We addressed this in L253-256, P. 14: "In the Barents-Kara region, SIC variability is closely linked to Q300 and the seasonal peak of correlations occurs in the freezing season (Jan-Feb), consistent with the timing of highest AW inflow towards the Arctic (e.g., Beszczynska-Möller et al., 2012)"

*Reviewer's comment*: Figure 6. As said in text, the empty boxes represent non-significant correlations (line 223). That information is better in the figure caption.

**Response**: We addressed this in Figure 6 caption, P.15: "[...] All values shown in color are significant at the 95\% level; empty boxes denote non-significant correlation [...]"

*Reviewer's comment*: Line 252. This process should be explained more in detail in terms of physics. How is the SIC prediction skill reduction associated with Qml during the earlier period and the MLD shoaling during the later period? Figure 7 presents a region of positive correlation in the Beaufort Sea in 2001-2021 that looks interesting. What physics explains that and why does it emerge in 2001-2021 compared to 1980-2000?

**Response**: We reworded L284-286, P.14 as follows: "Interestingly, the spatial pattern of correlation in the northeastern Beaufort Sea indicates a reduction of the area of SIC prediction skill associated with Qml during 2001-2021 (Fig. 7c,d). This area corresponds to the region of MLD shoaling during 1980-2021 (Fig. 2b)". We do not imply causality between the reduction in the Q-SIC correlation in the northeastern Beaufort Sea and MLD shoaling. Rather, we wish to point out that the region of reduced prediction skill corresponds to an area of MLD shoaling between 1980-2021 (Figure 2b). In this area of the Beaufort Sea, the correlation between Qml(oct) and SIC(nov) in 2001-2021 is not significant in either reanalysis (95% confidence). Therefore, we cannot speculate that a specific physical process may be behind the reduced influence of Qml on SIC in the later period, as the statistical robustness of the correlation over these grid points may be affected by the very shallow mixed layer.

*Reviewer's comment*: Line 259. the statistically non-significant OHT trend. Supposedly you think that the positive OHT trend is physically significant because it is mentioned, right?

**Response**: We reworded lines 292-295, P.16: "[...] This is consistent with the statistically non-significant OHT trend in the Chukchi Sea (Fig. 5c) and the weak increase in Bering Strait inflow from 1980-2000 to 2001-2021 (Table 2)."

*Reviewer's comment*: Line 302. Did you calculate lag correlations longer than 3 three months? Would be interesting to see how correlations decrease with lag time providing information on characteristic time scales.

**Response**: we appreciate the suggestion to look into this. At timescales longer than sub-seasonal, there are larger regional and seasonal differences in the preconditioning role of ocean heat content in sea ice variability. Additionally, when raising the lag time of correlations above 3 months, the two products begin to diverge in some months. For this reason, and to maintain a sub-seasonal focus in the study, we considered 3 months as the maximum lag time in our analysis. Interestingly, correlations between SIC and Q in the Barents/Kara region remain significant from January to at least May up to a lag time of 5 months (ocean leading) in both C-GLORSv5 and ORAS5 (Fig. **R4** attached shows lag = 4 and lag = 5 correlations for C-GLORSv5). In the second period (2001-2021), SIC anomalies between Nov-May are significantly correlated with Q300 leading by up to 5 months. However, in the Laptev/E.Siberian region, the prediction skill of Q begins to decrease at lag = 4 months in both reanalyses, with the exception for some of the summer months in the second period (2001-2021). Also, larger discrepancies between the two reanalysis products emerge from lag = 4 months, suggesting increased uncertainty at timescales longer than sub-seasonal (not shown).

[Figure]

**R4**: As in Fig. 6 in the original manuscript, but showing lagged correlations between regional SIC and Q(Q300 and Qml), as well as lagged SIC autocorrelations (c-d, g-h panels), for lags of 4 and 5 months. Data are from the C-GLORSv5 reanalysis.

*Reviewer's comment*: *Lines 309-312. This explanation is plausible, safe and presented multiple times before (for example Fox-Kemper et al. 2021, Chapter 9.3.1.1). It is regrettably also almost the only regional physical mechanism presented in the paper.*

**Response**: L351-254, P.20 refers specifically to the stronger correlation between Qml and SIC in the Laptev/East Siberian region during 2001-2021 relative to 1980-2001 (Fig. 7 in the manuscript). To the best of our knowledge, this result has not been presented or discussed before, despite the fact that the ice-albedo mechanism has indeed been extensively studied. We expanded and clarified this text in L349-351.

*Reviewer's comment*: *Line 324. The authors have decided to exclude stratification and halocline, although they form an essential component to understand the results.*

**Response**: We agree that the strength of stratification and halocline stability are important parameters to understand changes in the ocean-sea ice system. However, we believe these aspects are better investigated in a set of model experiments aimed at assessing the response of sea ice to increased or reduced vertical stratification or mixing, following for example the approach used in Liang and Losch (2018) or Hordoir et al. (2022). We hope to be able to address these open questions in future research.

Liang, X., & Losch, M. (2018). On the effects of increased vertical mixing on the Arctic Ocean and sea ice. Journal of Geophysical Research: Oceans, 123(12), 9266-9282.

Hordoir, R., Skagseth, Ø., Ingvaldsen, R. B., Sandø, A. B., Löptien, U., Dietze, H., ... & Lind, S. (2022). Changes in Arctic stratification and mixed layer depth cycle: A modeling analysis. Journal of Geophysical Research: Oceans, 127(1), e2021JC017270.

**Response to Anonymous Reviewer 2**

**General Response:**

We thank the reviewer for providing insightful comments to help us improve the manuscript. We addressed the suggestions provided by adding clarifications in the text, and, where appropriate, by editing figures and tables. We hope our edits to the manuscript are satisfactory and our responses are thorough. What follows is a brief summary of our responses to the key points raised by the reviewer and how they were addressed. Point-by-point responses to specific comments are also included below.

1) Section 2.2 and 2.3: The presentation and description of equations 1, 2 and 3 were improved by providing additional details and clarifications, including a clearer explanation of the choice of reference temperatures.
2) Section 3.2 and Table 2: We recognize that our manuscript did not adequately account for the caveat of computing and comparing OHT through individual gateways relative to an arbitrary reference temperature. This issue was addressed by editing part of the text in this section and by replacing and updating parts of Table 2.
3) Section 4: We expanded our discussion of the additional links and underlying mechanisms for the patterns and correlations highlighted in our results, including the role of parameters that are not explicitly investigated in our work.

**Specific Comments**

*Reviewer's comment: Page 1 line 4: '...the role of ocean heat content... ...is poorly documented...'*
*I would rather not agree with this statement. In the recent decade, the role of ocean heat in sea ice variability has been addressed by a large and growing number of scientific papers and reviews, based both on observations and numerical models (many of them are referred to in this manuscript). While there are still many open questions, in particular about the detailed mechanisms of heat transfer, this topic is not as 'poorly documented' as the abstract suggests.*

**Response**: We generally agree that the role of ocean heat in sea ice variability has been widely explored. However, the majority of studies have focused on ocean heat transport rather than ocean heat content and on longer timescales than those we explore in our study. To address the referee's comment, we rephrased L3-4 in the Abstract to highlight the focus on sub-seasonal timescales of sea ice variability: "Yet, the role of ocean heat content in modulating Arctic sea ice variability at sub-seasonal timescales is still poorly documented."

*Reviewer's comment: Page 1 line 20: '...temperature fluctuations (Olonscheck et al., 2019) ...'*
*What temperature fluctuations are meant here, ocean or air? Olonscheck et al. paper is focused on atmospheric temperature fluctuations only.*

**Response**: Changed to "atmospheric temperature fluctuations" (L20, P.1)

*Reviewer's comment: Page 2 line 25: 'The Atlantic sector...'*
*What is precisely meant here as the Atlantic sector? The AW inflow should be described first and followed by detailing its impact of the Arctic Ocean (not the other way around).*

**Response**: This definition was incomplete. We revised this sentence by specifying the regions included in the Atlantic Arctic: "(including the Greenland, Barents, Kara and Laptev seas)" (L26, P.2) and in the Pacific Arctic: "Including the Beaufort, Chukchi and East Siberian seas") (L43).

*Reviewer's comment: Page 2 line 32: 'In these regions...'*
*What are 'these regions'? Polyakov et al. papers addressing changes in stratification and weakening of CHL are focused on the eastern Eurasian Basin.*

**Response**: We reworded this sentence (L30-34): "The combined effects of increased volume inflow (Smedsrud et al., 2022) and warmer temperatures (Wang et al., 2019) of the AW current have been linked to sea ice decline in the Barents Sea (e.g., Årthun et al., 2012; Smedsrud et al., 2013) and more recently in the eastern Eurasian Basin (Polyakov et al., 2020b). In the latter, the interaction between AW and sea ice has also been examined in relation to the weakening of the stratified, cold halocline layer and the consequent increase in vertical heat fluxes toward the surface (Polyakov et al., 2010, 2017)."

*Reviewer's comment: Page 2 line 41-42: '...a modest increase in observed temperature (Woodgate, 2018).'. In the more recent paper, Woodgate et al. (GRL, 2021) indicate a significant (not 'modest') warming of the Pacific inflow through Bering Strait.*

**Response**: We appreciate the suggestion. This sentence was reworded to avoid confusion arising from the term 'modest' (L44-48, P.2): "The Pacific sector of the Arctic (including the Beaufort, Chukchi and East Siberian seas) has also been the epicenter of some of the most remarkable episodes of sea ice loss in recent years (Comiso et al., 2017). While small in comparison to the inflow through Fram Strait (about 10 times smaller in volume and with a heat flux that is 1/3 of the Fram Strait heat flux; Woodgate et al. 2012), the inflow of Pacific Water (PW) through Bering Strait has also shown a warming trend that can accelerate sea ice loss (MacKinnon et al., 2021; A Woodgate and Peralta-Ferriz, 2021) and act as a trigger for early melt onset (Woodgate et al., 2010)."

*Reviewer's comment: Page 2 line 48: 'temperature increase of Arctic Ocean...'*
*Should be 'increase in the Arctic Ocean'.*

**Response**: Corrected in L53, P.2.

*Reviewer's comment: Page 3 lines 68-70: 'In this work, we use two eddy permitting global ocean reanalyses...'. While the details of two reanalyses selected for this study are described in the following section, there is no explanation provided to justify the choice. Why particularly these two products have been singled out for analyzing the impact of the upper OHC on regional sea ice variability? Both reanalyses are based on the same ocean model (albeit different versions), forced by ERAI, and have similar horizontal and vertical resolution. If the choice was motivated by a difference in assimilated observations and assimilation schemes in both products, it should be better clarified. The choice of the SIC product from ERA5 for correlation analysis could also deserve a comment.*

**Response**: The motivation for choosing C-GLORSv5 and ORAS5 was explained in the following paragraph (L86-89, P.3): "[...] Their extensive temporal coverage (1980-present) and their high horizontal and vertical resolution, which allows for a high level of detail in the surface layers, make these products particularly suited for the study of the polar upper ocean."

As far as the choice of ERA5 SIC data is concerned, this is motivated by the intent to have as close an estimate to observations as possible. Using SIC as a metric effectively reduces the uncertainty associated with measurements, as opposed to, for example, using sea ice thickness. In terms of data sources, ECMWF ERA5 data provide uniformly gridded, easily accessible estimates of this quantity, and this is why this product is chosen over others.

***Reviewer's comment***: *Page 3 line 75: '...their performance in polar regions has been widely evaluated with encouraging results...'*
*This statement leaves impression that two selected reanalyses (C-GLORSv5 and ORAS5) have been widely evaluated but references (papers) listed in the same line are focused on other models.*

**Response**: These lines have been reworded to clarify that the first paragraph (L82-85, P.3) refers to the performance of ocean reanalyses in general, while the second paragraph (L85-89) refers specifically to the products employed in our study.

"Furthermore, the performance of different reanalysis products in polar regions has been widely evaluated with encouraging results (e.g., Iovino et al., 2022; Lien et al., 2016; Ilıcak et al., 2016). With respect to fully modeled data, errors and uncertainties in the representation of ocean variables in reanalyses are effectively reduced through the assimilation of…".

[...]

"The CMCC C-GLORSv5 and ECMWF ORAS5 products specifically have also been employed in several applications (e.g. Takahashi et al., 2021; Carton et al., 2019) including in polar regions (e.g., Mayer et al., 2019; Shu et al., 2021, Nie et al. 2022). "

***Reviewer's comment***: *Page 4 Fig 1: I suggest indicating line colors for individual gateways in the caption.*

**Response**: Thank you for your suggestion, we addressed this in Fig. 1 caption (p.6)

***Reviewer's comment***: *Page 5-6, line 116-122: The equation and description of the ocean heat budget terms are very imprecise. Why is the budget named 'the ocean's surface heat budget' when it describes changes of ocean heat content across all boundaries (not only the surface)? The equation is given for some undefined 'vertical section' and some undefined 'area' without defining a control volume within which the heat content is calculated (and mass and salt are conserved). The Eq. 1 is true only if area S represents the entire vertical boundary and area A the entire surface of the control volume, what is not mentioned in the text. Qdiff is lateral heat diffusion (indeed negligible compared to advective term and surface flux) but can result from both diapycnal mixing and lateral turbulent mixing.*

**Response**: We agree that the terms of equation 1 could have been explained with more precision and we edited the text according to the referee's comments.

- L121, P.5 and elsewhere in the text the erroneous "surface heat budget" was changed to "ocean heat budget"
- Additionally, L125-127, P.5 have been reworded to specify the definition of a control volume: "Considering a control ocean volume with surface A and vertical section S, where mass and salinity are conserved, the heat budget is given by the balance between advective, vertical and diffusive heat flux terms (1) [...]"

- L132 and L147: "diffusive heat transport" has been changed to "lateral heat diffusion"

***Reviewer's comment***: *Page 6 line 125 and Equation 2: 'The net ocean heat transport into the Arctic is computed along each section...'*
*In Eq. 2 T should be explicitly a difference T-Tref since heat transport always is calculated relative to the reference temperature.*

**Response**: The term (T-Tref) was explicitly added to Equation 2.

***Reviewer's comment***: *While total advective OHT (ocean heat transport convergence integrated over the whole Arctic Ocean lateral boundary or a boundary of each defined region) is independent of a reference temperature, it is not true for the partial cross-section with the nonzero net volume flux. Using any arbitrary reference temperature makes a heat transport across a partial section arbitrary, therefore the contributions across different sections have no physical meaning and cannot be compared.*

**Response**: We acknowledged this caveat in L141-144 (A more detailed description of the edits in response to this comment can be found in the responses below).

***Reviewer's comment***: *Page 6 line 144: '...estimate ocean heat content...'*
*The quantity calculated using Eq. 3 is ocean heat content per unit area, not the ocean heat content.*

**Response**: Corrected in L158, P.7.

***Reviewer's comment***: *Page 7 Equation 3: Since only trends and anomalies of the ocean heat content are used in the following analysis, they are independent of any reference temperature. Still, it is confusing when the authors introduce different reference temperatures for calculating ocean heat content and in the previous section for calculating the ocean heat transport. I would suggest removing these ambiguities or providing a strong justification for using different Tref.*

**Response**: We removed this ambiguity by clarifying the reason for choosing different reference temperatures (edits are described in the bullet points below). The choice of $T_{ref} = 0°C$ in the calculation of OHT (equation 2) is justified by the intent to provide estimates of OHT that can be widely compared with those existing in the literature. Several studies have adopted this reference value in the past (e.g., Årthun et al., 2012; 2019; Lique and Steele, 2013, Smedsrud et al 2021). Similarly, the choice of $T_{freeze} = -1.8°C$ in the calculation of OHC (equation 3) is motivated by the intent to conform to existing estimates in the literature (Mayer et al. 2019), where Q is typically computed by removing the constant of freezing temperature. Furthermore, using a reference temperature of 0°C instead of -1.8°C in equation 3 (as done in equation 2) would yield negative values of Q (for instance in Fig 3, which shows time series of Qml and Q300), which seems physically inaccurate given that Q represents energy. As the referee pointed out, the quantities analyzed and discussed in the study (trends and anomalies) are independent of any reference temperature, therefore our choice does not affect the results but seems to conform better with the literature and make the interpretation of Q more intuitive.

To address the referee's comment, we added the following clarifications to the text:

- in Equation 3 on Page 7, Tref was renamed Tfreeze, to avoid confusion with the previous equation

- The choice of Tfreeze = -1.8°C in Equation 3 was motivated in L164-165: "Here, we adopt the constant of freezing as reference temperature following previous studies (e.g., Mayer et al., 2019; Korhonen et al., 2013) to obtain the heat available for potential sea ice melt."

Årthun, M., Eldevik, T., Smedsrud, L. H., Skagseth, Ø., & Ingvaldsen, R. B. (2012). Quantifying the influence of Atlantic heat on Barents Sea ice variability and retreat. Journal of Climate, 25(13), 4736-4743.

Årthun, M., Eldevik, T., & Smedsrud, L. H. (2019). The role of Atlantic heat transport in future Arctic winter sea ice loss. Journal of Climate, 32(11), 3327-3341.

Lique, C., & Steele, M. (2013). Seasonal to decadal variability of Arctic Ocean heat content: A model‑based analysis and implications for autonomous observing systems. Journal of Geophysical Research: Oceans, 118(4), 1673-1695.

Mayer, M., Tietsche, S., Haimberger, L., Tsubouchi, T., Mayer, J., & Zuo, H. (2019). An improved estimate of the coupled Arctic energy budget. Journal of Climate, 32(22), 7915-7934.

*Reviewer's comment*: *Page 7 lines 156-159: description of anomaly time series and the lagged correlations. It is obvious (albeit not explicitly said) that grid-point time series of ocean heat content per unit area and grid-point values of different parameters are used to construct maps of annual trends on Fig. 2. However, it is not clear how was the lagged correlation calculated. Is the correlation calculated between anomalies of ocean heat content (integrated over each region or entire Arctic Ocean) and region-averaged anomalies of SIC or between region-averaged anomalies of ocean heat content per unit area and region-averaged anomalies of SIC? It will not change the result but should be better explained. Why not to simply use the ocean heat content (in J) for Qml and Q300 for the selected regions and the entire Arctic Ocean instead the less intuitive region-averaged ocean heat content per unit area (in J m-2) everywhere except Fig. 2 showing spatial trends?*

**Response**: We addressed the referee's suggestion with the following edits:

- in L173-175, P.7, we clarified that the lagged correlations are calculated between region-averaged anomalies of Q and SIC: "To investigate how upper ocean heat content at both target depths influences sea ice variability on sub-seasonal time scale, lagged correlations between region-averaged anomalies of Q and SIC are computed for each month, with the ocean leading sea ice by a lead time of one to three months."
- Regarding the unit of Qml and Q300, we first introduce this quantity in Figure 2, which shows spatial trends and therefore uses J/m2. For consistency, we keep the same unit in Figure 3, which shows region-averaged time series of Qml and Q300. This is mainly to avoid confusion in introducing a different unit (area-integrated values in J). Additionally, the two reanalyses are produced with the same grid so converting the values to J does not provide any added value to their comparison.

*Reviewer's comment*: *Page 7 line 168: 'positive SSHF trend... ...suggests reduced ocean cooling...' Since Fig. 2 shows annual trends, it is difficult to distinguish if the positive trend results from a reduced cooling in winter or an increased warming in summer. Only seasonal trends could show which process dominates.*

**Response**: We agree with the referee. To address this issue, we replotted Fig. 2 (L182-183, P.8) so that it now shows SSHF in separate seasons (MJJ and OND). This helps to distinguish the processes behind the trends depicted. L185-189, P.9 describe the seasonal trends in SSHF.

**Reviewer's comment**: *Page 9 lines 175-183: This paragraph discusses trends in different times series, based on Fig. 3 which do not depict any lines representing trends.*

**Response**: Note: Fig. 3 (P.11) was updated to show absolute values of SIC and Q instead of anomalies. A reference to trend values, where relevant, was added to the text in L198-202, P.9. Unfortunately, citing all trend values or annotating them in the Fig.3 panels would make the figure and text less legible (since there are three variables, two data products and 5 regions).

**Reviewer's comment**: *Page 9 line 183: '...except for a minor difference... ...at the beginning of the time series.'. There is also a significant difference between C-GLORSv5 and ORAS5 for the Laptev and East Siberian region in the last decade. Any idea what could be a reason?*

**Response**: When plotting absolute values of ocean heat content (see the revised version of Fig. 3, P.11), the difference between the two reanalyses in the Laptev-E.Siberian region appears negligible, since the two products agree well on the interannual variability as well as the trend. ORAS5 simulates slightly lower potential temperature than C-GLORSv5, with the exception for the last period (approx. 2015-2020). While it is difficult to identify the reason for this difference, it is not surprising given that the two products assimilate different observations. We mentioned this aspect in L203-205, P.9.

**Reviewer's comment**: *Page 9-12 Section 3.2 and Table 2.*
*A comparison of ocean heat transport across partial section is simply wrong because not only the absolute values but also their proportions will change with changing the reference temperature. A detailed explanation is provided in the paper by Schauer and Beszczynska-Möller (2009) which is among the listed references. Tsubouchi et al. (2018), Årthun et al. (2016), or Lique et al. (2013) also discuss limitations arising from the arbitrary choice of a reference temperature when calculating ocean heat transport across partial sections with nonzero net volume flux - all these papers are cited in the manuscript, but the authors have somehow decided to entirely ignore the problem. As a consequence, all numbers describing OHT across the four main gateways and Fig. 4a are physically meaningless since their ratio (and also ratio between their numbers in two reanalyses) will change with different reference temperature. In particular, the lower part of Table 2 is erroneous as the authors compare partial heat transport through individual gateways which in each case was calculated with a different reference temperature.*

**Response**: we addressed the referee's comment with the following edits to the text and to Table 2 (P.10).

- We added a paragraph acknowledging this caveat and urging caution in the interpretation of heat transports through different gateways (L141-144, P.6): "It is widely recognised that heat transports are sensitive to the choice of reference temperatures. Therefore, the reader should note that when considering partial sections, the estimated values of OHT are always dependent on the arbitrary choice of reference temperature, since there is a non-zero net volume flux (i.e., the volume transport across individual sections is not balanced; see Schauer and Beszczynska-Möller, 2009)"
- Table 2 (p.10): We edited this table so that only existing estimates of OHT that are referenced to 0C are included for comparison. We also stated this clearly in the table caption.

- In the paragraph on L220-226, P.11, we edited the text to specify that comparisons with the literature are limited to studies using the same reference temperature, consistent with the updated version of Table 2.

*Reviewer's comment: The majority of following discussion and conclusions is based on times series of ocean heat transport/content in the entire Arctic Ocean or ocean heat content in the individual regions so that the OHT time series across individual gateways in Section 3.2 are used mainly for a comparison between two reanalyses. Therefore, I would strongly suggest omitting the questionable partial heat transports and focus on the total OHT in the Arctic Ocean for assessing the budget closure. Alternatively, more comprehensive discussion of errors arising from the arbitrary choice of a reference temperature will have to be included (some examples can be found in the papers listed above). However, a comparison of OHT calculated with different reference temperatures across partial sections will always be a serious mistake.*

**Response**: We appreciate the suggestion. While we agree that this caveat should be better and more explicitly addressed in our manuscript (see edits below), our estimates for OHT through individual gateways in Fig. 4a and Table 2 are mainly aimed at making sure that there is a satisfactory agreement between the two reanalyses and that their sum (total OHT to the Arctic) is well balanced by surface fluxes (Fig 4b). Since the heat transports across straits in Fig. 4a are calculated using the same reference temperature, we can compare them with each other, as done in previous studies (e.g., Muilwijk et al. 2018). The main purpose of this section is therefore to establish that our data are robust and can be effectively used for the following stage of analysis (regional heat budgets). To address the referee's comment, we discussed the limitation of this approach in L141-144, P.6.

Muilwijk, M., Smedsrud, L. H., Ilicak, M., & Drange, H. (2018). Atlantic Water heat transport variability in the 20th century Arctic Ocean from a global ocean model and observations. Journal of Geophysical Research: Oceans, 123(11), 8159-8179.

*Reviewer's comment: Page 12 line 211: '...Net heat transport into the Kara-Barents Sea region has been increasing... ... the upward trend in BSO inflow.'*
*The BSO inflow was neither shown nor discussed in the paper. Please, either provide a relevant reference or comment on the BSO inflow variability in reanalyses.*

**Response**: The use of the term "inflow" was erroneous, we meant "an upward trend in OHT through the BSO". This was corrected in L237, P.12.

*Reviewer's comment: Page 12 line 220 and Fig.6: '...and ocean heat content...'*
*It should be 'ocean heat content anomalies'. The next two sentences repeat the Fig. 6 caption and can be removed. What are the white boxes in correlation plots (non-significant correlations? this can be deducted from the text but should be included in the caption).*

**Response**: we addressed the referee's comment with the following edits:

- "Ocean heat content" was changed to "ocean heat content anomalies" in L247, P.13.
- Lines repeating the Fig. 6 caption (P.15) were deleted, and we added in the figure caption that empty boxes denote non-significant correlations: "All values shown in color are significant at the 95% level; empty boxes denote non-significant correlations".

*Reviewer's comment*: Page 12 lines 226-227: '...the seasonal peak of correlation... ...in the freezing period, when AW inflow is at its annual maximum.'
*As mentioned above, the paper does not address variability of the AW inflow so either a proper reference is needed or additional comment on the BSO transport as reproduced by reanalyses. With AW inflow at its maximum, do the authors mean the strongest inflow, the warmest AW or both (or simply the increased ocean heat influx)?*

**Response**: We reworded this sentence and added an appropriate reference (L253-256, P.13): "In the Barents-Kara region, SIC variability is closely linked to Q300 and the seasonal peak of correlations occurs in the freezing season (Jan-Feb), consistent with the timing of strongest AW inflow towards the Arctic (e.g., Beszczynska-Möller et al., 2012)."

*Reviewer's comment*: Page 12 line 229: '...is stronger and shifted towards the autumn...'
*While the correlation is indeed stronger in the later period, a shift towards the autumn is not so obvious (the maximum correlation is still in the same month in the Laptev-East Siberian Sea region).*

**Response**: This is true for Q300, but not for Qml: the correlation between Qml and SIC in the Laptev-East Siberian region shifts from August to October/November between the two periods. However we agree with the referee that the text was unclear, so we reworded it in L258-260, P.14 "The 1-month lagged Q-SIC correlation is increased, particularly in the autumn (note the shift in the maximum Qml-SIC correlation from August to October-November in the Laptev-East Siberian region). "

*Reviewer's comment*: Page 12-13 Section 3.3: What about correlations between SIC and Q300 for the other two regions and their changes between two periods? Even if they are less pronounced, it would be worth to mention the patterns.*

**Response**: Thank you for your suggestion. We attach a figure showing 1-month lag correlations in the Chukchi and Beaufort seas for C-GLORSv5 (Fig. **R1)**. We included a paragraph in Section 3.3 that briefly summarizes the changes in Q-SIC correlations between the two periods in the Chukchi and Beaufort seas (L264-266, P.14): "[...] Changes in the intensity and seasonality of the Q-SIC coupling between the two periods are also noticeable in the Chukchi Sea, where the correlation between SIC and Q anomalies at both target depths appears intensified in the autumn months (Oct-Dec) during 2001-2021 (not shown). No substantial changes are observed in the Beaufort Sea (not shown)."

*Reviewer's comment*: Page 13 line 241: '...of the Pacific Arctic (Laptev-East Siberian, Chukchi, and Beaufort seas) ...'. While the East Siberian Sea is located in the Pacific Arctic, the Laptev Sea belongs rather to the European Arctic.*

**Response**: Corrected in L275, P.14

*Reviewer's comment*: Page 14 line 248: '...than the October to November SIC auto-correlation.'
Please add '(not shown') for the latter.*

**Response**: Added in L281, P.14

*Reviewer's comment*: Page 14 line 250: '... a shift in the ice-ocean coupling from summer to autumn'.
It is clearly visible that correlation in autumn is stronger in the Laptev-East Siberian Sea in the latter*

*period than before. But how do we know that in the earlier period the correlation there was stronger in summer? Can it be seen from Fig. 6?*

**Response**: The wording of this sentence was misleading; indeed the statement refers to Fig. 6 and the text has been reworded as follows (L281-284, P.14): "While this is true for both periods, there is a noticeable increase in the strength of the correlation in the Laptev-East Siberian region during 2001-2021, consistent with the shift in the timing of maximum ice-ocean coupling from summer to autumn (Figure 6 b-f)."

[Figure]

Fig. R1: As in Figure 6 in the manuscript, but showing lag 1 month Q-SIC correlations and SIC auto-correlations in the Chukchi and Beaufort Sea for the C-GLORSv5 reanalysis.

*Reviewer's comment: Page 15 lines 255-259: '...greater difference in predictability... ...in the Laptev-East Siberian region... ...modest increase in Bering Strait inflow...'*
*What is the physical mechanism behind an increase of predictability in the region less influenced by the Pacific-origin ocean heat (Laptev-East Siberian) as compared to the Chukchi Sea? Intuitively, predictability based on ocean heat should be strengthened where warming signal (heat surplus) becomes stronger.*

**Response**: Yes, indeed that was our initial intuition, too. However, it appears that the increase in predictability in this region is mainly due to the change in SSHF (as it can be seen from Figure 5g, which shows a warming trend since the 2000s associated with reduced ocean heat loss and sea ice decline). The warming signal has indeed become stronger in the last two decades, but we find no evidence that this is due to an increase in OHT from the Pacific Ocean. One of our main conclusions is that it is the change in surface fluxes that appears to drive a large part of the strengthening in Q-SIC coupling, and the Laptev-East Siberian region makes a particularly strong case for this argument. We discuss possible reasons for this in L340-353, P.20, where we cite evidence from the recent study by Sumata et al. (2023). In this study, the authors provide a detailed explanation of the processes that caused a shift to warmer/ice-depleted conditions in this region from the 2000s, which is in good agreement with what our results revealed.

To address the referee's comment, we reworded this sentence (L289-293, page 15): "However, we find a greater difference in the predictability of November SIC between the first and second period in the Laptev-East Siberian region (from r=-0.52 to r=-0.86 in C-GLORSv5; from r=-0.51 to r=-0.78 in ORAS5) than in regions of greater PW influence, i.e., the Chukchi Sea (no change in C-GLORsv5; from r=-0.87 to r=-0.88 in ORAS5). This is consistent with the statistically non-significant OHT trend in the Chukchi Sea (Fig. 5c) and the weak increase in Bering Strait inflow from 1980-2000 to 2001-2021 (Table 2)."

Sumata, H., de Steur, L., Divine, D. V., Granskog, M. A., & Gerland, S. (2023). Regime shift in Arctic Ocean sea ice thickness. Nature, 615(7952), 443-449.

*Reviewer's comment: Page 15 lins 264-265: 'December Qml and Q300 anomalies... ...correlation is higher for Q300 (Fig. 8)'. Qml is not shown on Fig. 8.*

**Response**: Corrected in L298, P.16: "December Qml (not shown) and Q300 anomalies are both strongly anticorrelated with January SIC anomalies, although the correlation is higher for Q300 (Fig. 8). "

*Reviewer's comment: Page 15 line 266-267: 'An intensification of the Q300 link... ...large part of the marginal ice zone in proximity of the St. Anna Trough.' Is a stronger correlation near MIZ in the recent period only the result of the farther northward advection of warmer Atlantic water (and less ice formation in winter) or can other physical mechanisms also play significant role in making this link stronger (e.g. thinner and more mobile ice in MIZ, local wind forcing, other factors)?*

**Response**: Based on our analysis, we cannot speculate on which specific mechanisms contribute to making this link stronger. However, it is likely that a reduction in sea ice thickness and age, particularly after 2007 (Sumata et al. 2023 discuss this shift in detail), played a central role in the heightened surface heating and, consequently, the stronger correlation with ocean heat content. Thinner and more mobile ice conditions contribute to creating areas of open ocean and make it difficult for sea ice to survive into the following year, particularly in the presence of strong wind forcing.  Feedbacks between sea ice thickness, mobility/residence time and upper ocean heating can all contribute to an overall stronger coupling between sea ice and ocean heat content, though addressing their relative contributions would require a more detailed study of specific years/episodes in the historical record (e.g., the winters of 2012 and 2013, which were characterized by the prolonged presence of polynyas north of Svalbard, Ivanov et al. 2016).

Sumata, H., de Steur, L., Divine, D. V., Granskog, M. A., & Gerland, S. (2023). Regime shift in Arctic Ocean sea ice thickness. Nature, 615(7952), 443-449.

Ivanov, V., Alexeev, V., Koldunov, N. V., Repina, I., Sandø, A. B., Smedsrud, L. H., & Smirnov, A. (2016). Arctic Ocean heat impact on regional ice decay: A suggested positive feedback. Journal of Physical Oceanography, 46(5), 1437-1456.

***Reviewer's comment***: *Page 15 line 273: '... predictability is maintained... ...up to lead time of 3 months in both reanalyses...'*
*While a maximum shown time lag of 3 months supports the role of ocean heat content as a predictor of sea ice in the Barents-Kara Sea, I am curious whether the correlation will decrease with increasing time lag, and when the correlation is at the maximum (from Figs 6, 9 and Appendices, in some months and regions correlation actually increases with a larger time lag). E.g. Schlichtholz (Sci Rep, 2019, doi 10.1038/s41598-019-49965-6) showed that the maximum correlation between Atlantic water temperature in the BSO inflow and sea ice area in the Barents Sea was found for the time lag of 5 months for the period after 2004. Onerheim et al. (2015) showed that including local (meridional) winds in the predictive framework significantly increased prediction skill on interannual time scale. How could the sub-seasonal predictability of SIC be improved by using other parameters in addition to the upper ocean heat content?*

- *"I am curious whether the correlation will decrease with increasing time lag, and when the correlation is at the maximum"*

  **Response**: Thank you for your suggestion. Interestingly, correlations between SIC and Q in the Barents/Kara region remain significant from January to at least May up to a lag time of 5 months (ocean leading) in both C-GLORSv5 and ORAS5 (Fig. **R2** attached shows this for C-GLORSv5). In the second period (2001-2021), SIC anomalies between Nov-May are significantly correlated with Q300 leading by up to 5 months. However, in the Laptev/E.Siberian region, the prediction skill of Q begins to decrease at lag = 4 month in both reanalyses, with the exception for some of the summer months in the second period (2001-2021). Also, larger discrepancies between the two reanalysis products emerge from lag = 4 months (not shown), suggesting increased uncertainty at timescales longer than sub-seasonal. For this reason, and to maintain the sub-seasonal focus in our study, we limited the lag time of correlations to 3 months.

- *How could the sub-seasonal predictability of SIC be improved by using other parameters in addition to the upper ocean heat content?":*

  **Response**: We appreciate the suggestion to look into this in a follow-up study. On the timescales we consider, wind stress and SLP anomalies may indeed offer additional information for the predictability of the Q-SIC coupling based on their demonstrated effect on ocean heat content (e.g., Sandø et al 2010) and transport (Muilwijk et al 2018, Lien et al 2017). Addressing this question, however, requires a region-specific assessment of the key drivers of sea ice variability, given that different parameters will have very different relative influences in the four Arctic sub-regions.

Sandø, A. B., Nilsen, J. Ø., Gao, Y., & Lohmann, K. (2010). Importance of heat transport and local air‑sea heat fluxes for Barents Sea climate variability. Journal of Geophysical Research: Oceans, 115(C7).

Muilwijk, M., Smedsrud, L. H., Ilicak, M., & Drange, H. (2018). Atlantic Water heat transport variability in the 20th century Arctic Ocean from a global ocean model and observations. Journal of Geophysical Research: Oceans, 123(11), 8159-8179.

Lien, V. S., Schlichtholz, P., Skagseth, Ø., & Vikebø, F. B. (2017). Wind-driven Atlantic water flow as a direct mode for reduced Barents Sea ice cover. Journal of Climate, 30(2), 803-812.

[Figure]

**R2**: As in Fig. 6 in the original manuscript, but showing lagged correlations between regional SIC and Q(Q300 and Qml), as well as lagged SIC autocorrelations (c-d, g-h panels), for lags of 4 and 5 months. Data are from the C-GLORSv5 reanalysis.

*Reviewer's comment: Page 15 line 274: '... especially in months with inherently lower sea ice predictability.' The statement is rather unclear. What do the authors mean by months with inherently lower predictability? The months with lower or non-significant autocorrelation?*

**Response**: Yes, we reworded the sentence to make this statement clearer (L307-309, Page 17): "[...] Highlighting the importance of ocean heat content as a precursor of SIC variability in this region, especially in months with lower or non-significant SIC auto-correlation."

*Reviewer's comment: Page 15 line 276: '...shed light on the mechanisms underlying the regional patterns...'*
*While regional correlations and spatial patterns have been widely discussed in the paper, there is very limited explanation of potential physical mechanisms laying behind described statistical links.*

**Response**: Though we recognize that our manuscript provides limited insight into specific physical mechanisms, these cannot be thoroughly examined in a single paper, as each individual sub-region would require a detailed study in the context of its oceanic and atmospheric regime. In our view, this is better addressed in a separate study building on the key results highlighted in this manuscript.

To address this comment, we expanded the discussion of possible underlying mechanisms by placing our results in the context of existing literature.

- L332-339, P.18: discuss the role of large-scale weather patterns and in particular the NAO in the variability of ocean warming and sea ice in the Barents-Kara region
- L349-352, P.20: clarify the link between our results for the Laptev-East Siberian region and existing evidence of the role of the ice-albedo feedback
- L368-374, P.20: discuss the importance of other parameters and mechanisms that are not explicitly addressed in the study

- We reworded L311, P.17: "The analysis presented in this study aims to shed light on the implications of upper ocean warming for regional sea ice variability. "

*Reviewer's comment: Page 19 line 311: '...the expansion of Fram Strait influence along the Siberian shelf...'. What exactly do the authors mean as 'the expansion of Fram Strait influence'? When travelling along the Siberian shelf, Atlantic water originating from Fram Strait is already sheltered by CHL from the surface mixed layer. Shoaling of the AW layer in the Eastern Eurasian Basin (as shown by Polyakov et al. in a series of papers) not necessarily contributes to the ocean heat anomaly in the mixed layer (Qml) and its coupling to SIC.*

**Response**: We agree that the shoaling of the AW is not a sufficient condition for the warming of the mixed layer (and associated effects on sea ice melt). However, we cannot rule out this factor entirely, given that stratification and halocline stability, mixing and sea ice melt have become strongly interrelated over recent decades. For instance, there is evidence that a substantial amount of AW heat is able to reach the surface layer and contribute to ice melt in the WNB (Ivanov et al. 2012). Only a detailed analysis of the structure of water masses and strength of stratification along the Siberian shelf over our study period could provide insights into this aspect, and we hope to be able to look into it in future work.

Ivanov, V. V., Alexeev, V. A., Repina, I., Koldunov, N. V., & Smirnov, A. (2012). Tracing Atlantic Water signature in the Arctic sea ice cover east of Svalbard. Advances in Meteorology, 2012.

*Reviewer's comment: Page 19 lines 329-330: '... strong coupling between ocean heat content and sea ice anomalies... ...important implications for sub-seasonal prediction... ...practical value to stakeholder...'. While this statement is generally true, the practical use of ocean heat content as sea ice predictor on sub-seasonal time scale can be limited, given existing biases in available reanalyses and the fact that in the Arctic, ocean heat content is only very sparsely measured observable (e.g. due to the limited use of Argo floats as compared to the global network).*

**Response**: In our view, these limitations should not prevent these products/variables from being used. On the contrary, we think they should be explored more extensively in future work, given their demonstrated value for the predictability of sea ice. We hope that our results can further motivate the ongoing development of ocean reanalyses (among other active improvements, bias reduction and an extended temporal and spatial coverage are particularly crucial). Additionally, while sub-surface measurements are indeed still sparse, we hope our study can also contribute to making a strong case for why the expansion of the observational network deserves priority in the study of the Arctic climate.

*Reviewer's comment: Page 19 line 334: '...thou improved accuracy...'*
*Should be 'through'.*

**Response**: Corrected in L383, P.21.

**Response to Anonymous Reviewer 3**

**General response:**

We thank the reviewer for suggesting clarifications in our manuscript and motivating a more in-depth discussion of some issues. We have addressed the comments by editing and expanding parts of the text. We hope our responses are thorough and our improvements are satisfactory.

***Reviewer's comment:*** *What I am most concerned about is why the paper did not consider the effects of runoff and precipitation when calculating the heat budget of the upper ocean. Although these two contributions are likely not obvious, it is necessary to quantify them appropriately, otherwise it may appear incomplete. Especially for the assessment of ocean heat budget in the Laptev-East Siberian Seas, the role of runoff should still be quite evident during the warming seasons. If it is difficult to make quantitative evaluations, some qualitative discussions are also necessary.*

**Response:** In the NEMO ocean model, river runoff is a prescribed quantity that affects the Arctic ocean's mass budget but does not contribute to the ocean heat budget. The river discharge enters the ocean, inserted to the top model cell, assumed to be fresh (zero salinity) and at the model cell temperature. No temperature information is prescribed. Unfortunately, this limitation prevents us from quantitatively assessing the role of river runoff to our Arctic or regional heat budgets.

However, we agree that the role of runoff deserves further investigation, since there is evidence that increased river runoff arising from an intensified hydrological cycle can increase the stability of the upper ocean (Holland et al., 2007). This could in turn reduce vertical mixing, with important implications for sea ice. We believe that these aspects are best investigated with a more advanced modeling approach, for example following Nummelin et al. (2016).

Holland, M. M., Finnis, J., Barrett, A. P., & Serreze, M. C. (2007). Projected changes in Arctic Ocean freshwater budgets. Journal of Geophysical Research: Biogeosciences, 112(G4).

Nummelin, A., Ilicak, M., Li, C., & Smedsrud, L. H. (2016). Consequences of future increased Arctic runoff on Arctic Ocean stratification, circulation, and sea ice cover. Journal of Geophysical Research: Oceans, 121(1), 617-637.

***Reviewer's comment:*** *The influence of "the weakening of the stratified and cold halocline layer" on the ocean heat content and vertical heat fluxes towards the surface: It should be said that it is not a clear question. Based on your studies, you should be able to partially answer this question to make it clearer. Therefore, I suggest that you further discuss this issue.*

**Response**: In L368-374, P.20, we expanded our discussion of the open questions associated with changes in vertical stratification and halocline stability, drawing upon existing literature. Building on our results, we cannot provide quantitative insights into this issue, however we aim to include the analysis of halocline stability in the context of future work.

***Reviewer's comment:*** *Figure 1: Most of the heat transported northward through the Davis Strait may be consumed through ocean-coast interactions in the strait area and cannot enter the Arctic Ocean to participate in heat exchange between the atmosphere, sea ice, and ocean. This potential impact should be taken seriously.*

**Response**: For the computation of Davis Strait OHT, as well as its location in Fig. 1, we follow what is traditionally defined in the literature (e.g., Muilwijk et al 2018, Curry et al 2011). Indeed the contribution of Davis Strait OHT is small in comparison to the other gateways, however the closure of the heat budget in Section 3.2 (Figure 4b in the manuscript) demonstrates that the difference between the heat that enters Davis Strait and the portion of this heat that participates in the Arctic ocean heat budget (after ocean-coast interaction), if not null, must be negligible. Additionally, Davis Strait OHT is not actually employed in our regional heat budget equation, as none of the sub-regions in Fig. 1 are directly adjacent to Davis Strait. Thus, our analyses on sea ice variability are not quantitatively affected by the relative amounts of Davis Strait OHT that may be lost to ocean-coast interactions.

Muilwijk, M., Smedsrud, L. H., Ilicak, M., & Drange, H. (2018). Atlantic Water heat transport variability in the 20th century Arctic Ocean from a global ocean model and observations. Journal of Geophysical Research: Oceans, 123(11), 8159-8179.

Curry, B., Lee, C. M., Petrie, B., Moritz, R. E., & Kwok, R. (2014). Multiyear volume, liquid freshwater, and sea ice transports through Davis Strait, 2004–10. Journal of Physical Oceanography, 44(4), 1244-1266.

*Reviewer's comment: What is the cooling effect of the Barents Sea? What role will this effect play in the ocean heat budget of the Barents Sea? It needs to be explained in a more advanced context.*

**Response**: A more detailed explanation of the cooling effect of the Barents Sea is now provided in L37-42, P.2 in the Introduction: "The expansion of the Barents Sea open water region leads to enhanced heat release from the ocean to the colder atmosphere, a phenomenon known as the "Barents Sea cooling machine" (Skagseth et al., 2020). In this part of the Arctic, complex feedbacks exist between sea ice retreat, anomalous atmospheric circulation, and induced changes in ocean heat transport by wind-driven currents (e.g., Mohamed et al., 2022). Observations over the past 20 years have shown reduced heat loss in ice-free areas of the Barents Sea, which exceeds the increase driven by the sea-ice retreat (Skagseth et al., 2020)." We also discuss changes in SSHF in the Barents Sea as they emerge from our results (L184-189, P.9 and 322-330, P.17) and in the context of existing literature (L309-319).